# Modular RNA motifs for orthogonal phase separated compartments

Jaimie Marie Stewart[1,11], Shiyi Li [2,11], Anli A. Tang [3,11], Melissa Ann Klocke[3,10], Martin Vincent Gobry [4], Giacomo Fabrini [5,6,7], Lorenzo Di Michele [5,6,7], Paul W. K. Rothemund [1,8,9] ✉ & Elisa Franco [2,3] ✉

Recent discoveries in biology have highlighted the importance of protein and RNA-based condensates as an alternative to classical membrane-bound organelles. Here, we demonstrate the design of pure RNA condensates from nanostructured, star-shaped RNA motifs. We generate condensates using two different RNA nanostar architectures: multi-stranded nanostars whose binding interactions are programmed via linear overhangs, and single-stranded nanostars whose interactions are programmed via kissing loops. Through systematic sequence design, we demonstrate that both architectures can produce orthogonal (distinct and immiscible) condensates, which can be individually tracked via fluorogenic aptamers. We also show that aptamers make it possible to recruit peptides and proteins to the condensates with high specificity. Successful co-transcriptional formation of condensates from single-stranded nanostars suggests that they may be genetically encoded and produced in living cells. We provide a library of orthogonal RNA condensates that can be modularly customized and offer a route toward creating systems of functional artificial organelles for the task of compartmentalizing molecules and biochemical reactions.

The discovery of membrane-less organelles is transforming our understanding of cellular biology and disease[1]. These organelles, also known as biomolecular condensates, arise when mixtures of nucleic acids and proteins segregate into spatially separated phases due to specific and non-specific (electrostatic and hydrophobic) molecular interactions[2]. Condensation broadly describes the formation of viscoelastic aggregates, and in biology, it is typically associated with a phase transition of mixtures of both RNA and proteins[2,3] that is driven by the interaction of intrinsically disordered domains (IDRs) of proteins[4], protein-RNA interactions[5] or RNA-RNA interactions[6]. Distinct, immiscible condensates—here termed "orthogonal condensates"

— arise through variations of the chemical identity and interactions of the protein and RNA components from which the condensates are composed. The prevailing model is that orthogonal condensates enable the spatial and temporal control of chemical components and biochemical reactions. In general, the greater the number of orthogonal condensates that can be created, the greater the number of distinct biochemical functions that can be performed. In cells there are dozens of functionally distinct condensates[7], that are involved in diverse gene regulation processes[8], cellular stress[9], and neurodegenerative diseases such as Alzheimer's Disease and ALS[10]. The development of libraries of synthetic, orthogonal condensates would allow for

[1]Department of Computing and Mathematical Sciences, California Institute of Technology, Pasadena, CA, USA. [2]Department of Bioengineering, University of California, Los Angeles, CA, USA. [3]Department of Mechanical and Aerospace Engineering, University of California, Los Angeles, CA, USA. [4]Interdisciplinary Nanoscience Center (iNANO), Aarhus University, Aarhus, Denmark. [5]Department of Chemical Engineering and Biotechnology, University of Cambridge, Cambridge, UK. [6]Department of Chemistry, Molecular Sciences Research Hub, Imperial College London, London, UK. [7]fabriCELL, Molecular Sciences Research Hub, Imperial College London, London, UK. [8]Department of Bioengineering, California Institute of Technology, Pasadena, USA. [9]Department of Computation & Neural Systems, California Institute of Technology, Pasadena, USA. [10]Deceased: Melissa Klocke. [11]These authors contributed equally: Jaimie Marie Stewart, Shiyi Li, Anli A. Tang. ✉e-mail: pwkr@dna.caltech.edu; efranco@seas.ucla.edu

the engineering of coordinated systems of controllable micro-compartments with distinct functions matching the complexity of living systems.

As sequence-programmable biopolymers, proteins and RNA have all shown promise as building blocks of artificial condensates. A fundamental design principle used in these investigations has been to introduce weak, non-specific, homotypic interactions that are thought to be essential for phase separation. This was made possible by engineering proteins to include IDRs present in naturally condensing proteins like FUS and SUMO proteins[11–14]. Similarly, artificial RNA condensates have been demonstrated using long molecules featuring expanded repeats of short sequences, which are found in nuclear foci associated with neurological diseases[15,16]. While IDRs and short repeats introduce multivalency, they also introduce promiscuous molecular interactions, thereby limiting the possibility of building coexisting yet immiscible condensates with distinct identities and tunable properties. An alternative approach has been pioneered through nanostructured DNA motifs that form condensates thanks to localized interactions whose specificity is achieved by rationally designed base-pairing[17–19]. Star-shaped DNA motifs have successfully been used to build coexisting but immiscible orthogonal condensates with programmable phase transitions[17,19,20].

Taking inspiration from the success of DNA nanostar condensates, and using techniques from RNA nanotechnology[21,22] we have designed and synthesized both multi-stranded and single-stranded RNA motifs that phase separate into orthogonal condensates. With the aid of computational models predicting RNA-RNA interactions, we demonstrate a suite of star-shaped RNA motifs, or nanostars, that generate RNA-dense droplets thanks to designed base-pairing domains, either linear sticky-ends or kissing loops, at the tip of their arms (Fig. 1). As shown in DNA nanostar analogs, this strategy makes it possible to obtain phase transitions by controlling nanostar affinity (determined by sequence and length of the base-pairing domains) and valency (determined by the number of arms)[17,18,23]. We adapt RNA nanostars to include an array of RNA aptamer domains that bind to small molecules and peptides, producing an expandable library of modular motifs that yield condensates with the capacity to recruit and segregate client molecules. Finally, we show that single-stranded RNA nanostars fold and produce condensates co-transcriptionally[24], and

could serve as genetically encoded building blocks to make RNA organelles with desired biophysical features and with the capacity to concentrate molecular targets. This feature is immediately applicable to building organelles in synthetic cells, as demonstrated in a companion paper by Fabrini et al[25].

## Results

### Condensates emerge from rationally designed multi-stranded RNA motifs interacting via sticky-ends

We begin by demonstrating RNA nanostars assembled from multiple RNA strands, a design that takes inspiration from DNA nanostars interacting via linear sticky-ends[18] (Fig. 2a). In our designs, arm sequences are distinct, but each arm presents the same palindromic sticky-end that controls nanostar affinity. We built several four-arm variants, labeled 4m1, 4m2, etc., where "m" denotes the multi-stranded nature of the constructs, "4" is the number of arms, and the final number denotes the sticky-end variant. The sequences of 15-nucleotide (nt) arms and their sticky-ends were optimized using NUPACK[26] to prevent repeats and unwanted interactions. To improve flexibility, we included unpaired adenines at the nanostar core and at the base of the sticky-ends[27]. To produce condensates, RNA was first transcribed in vitro and spin-column purified. Denaturing Polyacrylamide Gel Electrophoresis (PAGE) confirms the production of transcripts of the expected length, and a small amount of truncated products (Supplementary Fig. 4). Purified RNA was suspended in 40 mM HEPES, 100 mM KCl, 500 mM NaCl, which we will refer to as "assembly buffer", and thermally treated by a melt and hold protocol (Fig. 2b). The initial heating phase at 70 °C denatures unspecific binding and releases any structures from kinetic trapping. Nanostars form during the temperature drop from 70 °C to 50 °C, and condensates form during the 12-h long hold at 50 °C. Native PAGE of annealed samples including at least two of the participating strands shows the formation of complexes that migrate slowly or do not enter the wells (Supplementary Fig. 5). Additional hold temperatures and buffer conditions were screened in the Supplementary Information file (Supplementary Figs. 1–3). Samples were stained with 1X SYBR gold for imaging.

We found that 4–6 nt long sticky-ends yield condensates of varying morphology, including round droplets, aggregates of slowly

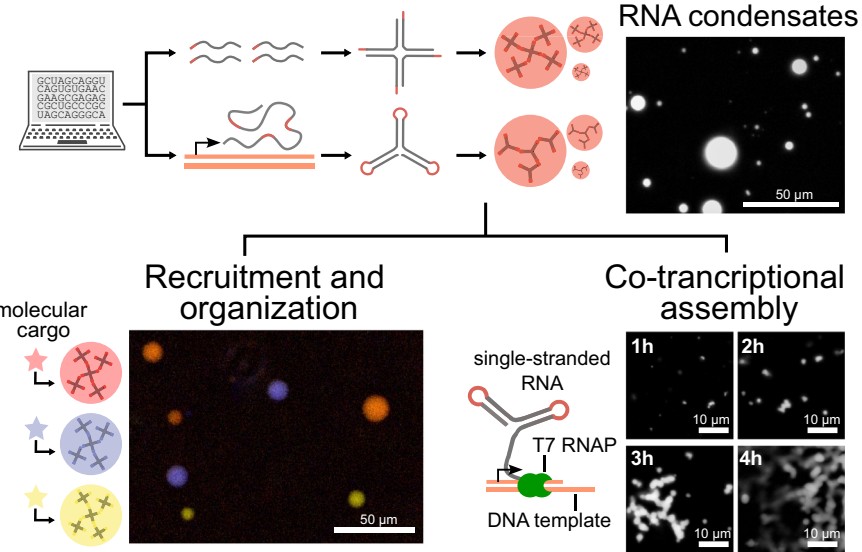

**Fig. 1 | Engineering synthetic RNA condensates for orthogonal separation, molecular recruitment, and co-transcriptional assembly.** We demonstrate the design and formation of RNA condensates from rationally designed multi-stranded and single-stranded RNA nanostars. These condensates have the capacity to recruit and organize molecules, and single-stranded motifs yield condensates co-transcriptionally. Scale bars: top right and bottom left, 50 μm; bottom right, 10 μm.

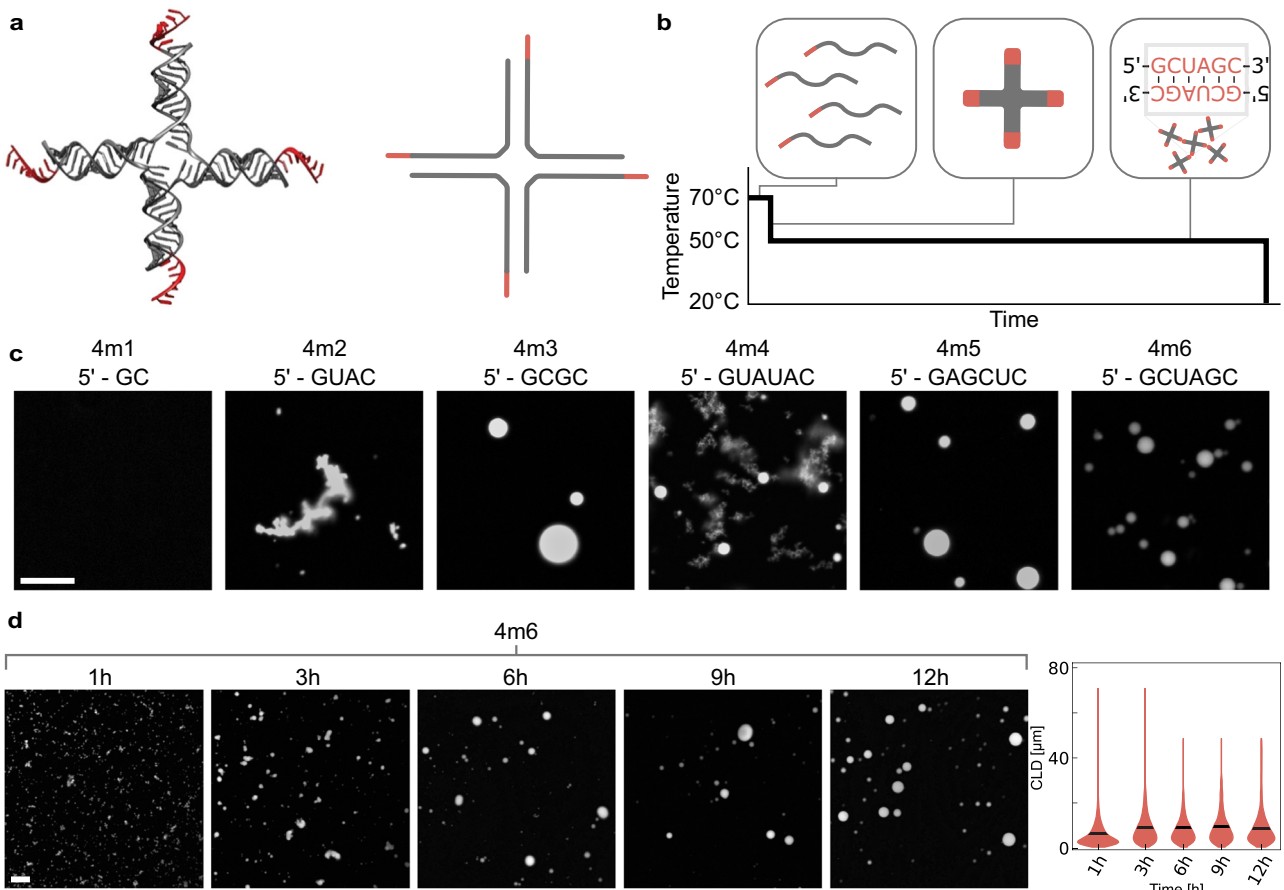

**Fig. 2 | Multi-stranded RNA nanostars yield condensates of variable size and morphology. a** PDB rendering and 2D representation of a multi-stranded RNA nanostar. **b** We use a "melt and hold" thermal annealing that sequentially promotes RNA denaturation (left), nanostar assembly (middle), and nanostar-nanostar interactions (right) that yield condensates. **c** Condensate morphology depends on the sequence and length of nanostar sticky-ends. **d** Growth of multi-stranded condensates (4m6) formed in 40 mM HEPES/100 mM KCl/500 mM NaCl and sampled for imaging over a 12-h incubation period. Right: condensate growth during incubation is confirmed by the image chord length distributions (CLD); black lines indicate the mean. Samples were stained with SYBR Gold and imaged. Images are representative of data collected in three replicates. Violin plots in (d) pool data from one sample, imaged 14 times. Scale bars: 40 µm.

fusing droplets, and "cloudy" aggregates (Fig. 2c). In contrast, no condensates formed in the absence of sticky-ends or with 2 nt long sticky-ends unless excess salt was added[27] (Fig. 2c and Supplementary Fig. 6). We observed the presence of aggregates in sticky-ends with high UA content. This can not be simply explained thermodynamically, given the higher free energy (ΔG) of the sticky-end (Supplementary Table 1); but likely due to the formation of UAUA or U quartet/tetrads[28–31]. We tested the influence of the cold temperature on variant 4m5, finding that lower hold temperatures (40–45 °C) yield aggregates, while temperatures in the range of 46–65 °C produce spherical assemblies; a 70 °C hold eliminates condensation (Supplementary Fig. 1). Condensates persist at high KCl concentrations (Supplementary Fig. 2), and the addition of MgCl$_2$ alters their melting temperature (Supplementary Fig. 3).

We tracked the condensation of variant 4m6 during the temperature hold, finding large condensates after 3–6 h of incubation (Fig. 2d). The average size of condensates increases during the hold step, as shown via chord length distribution (CLD) analysis[32–35] (Fig. 2d, right, and SI Methods 1.4). Chords are generated from binary masks of epifluorescence images, by measuring the intersections between straight lines and regions corresponding to condensates. CLDs are an expedient method to obtain information about the length scale of condensates regardless of their shape, which varies from spheres to aggregated networks depending on the nanostar design and assembly conditions. Fluorescence recovery after photobleaching (FRAP)

experiments show no recovery at room temperature, and minimal recovery at 50 °C (Supplementary Fig. 7), indicating gel-like behavior.

## Loop-loop interactions enable condensation of single-stranded RNA motifs

A requirement for potentially producing RNA nanostars in living cells is that they form at constant temperature. To achieve this, we developed RNA nanostars comprising a single strand rather than several distinct molecules. In these designs, sticky-ends are replaced by kissing loop (KL) domains placed at the end of consecutive stems that serve as nanostar arms (Fig. 3a). Like in the multi-stranded designs, the arm sequences are distinct to minimize misfolding but all KLs on a particular nanostar are identical. Given our goal of co-transcriptional formation, we decided to minimize transcript length and include only three consecutive arms, the lowest valency for condensation. Stem sequences were adapted from design 4m6, eliminating one of the arms. We started by testing the palindromic wild-type Human Immunodeficiency Virus (HIV) KL sequence (GCGCGC, variant termed 3sWT, where "s" denotes the single-stranded nature of the design and "3" the number of arms), and we also developed variants with non-palindromic sticky-ends (3sα-ζ), each including 2 distinct nanostars (Fig. 3b). First, we characterized the condensation of our designs using RNA transcribed by T7 polymerase, spin-column purified, and annealed with our melt and hold protocol in our assembly buffer.

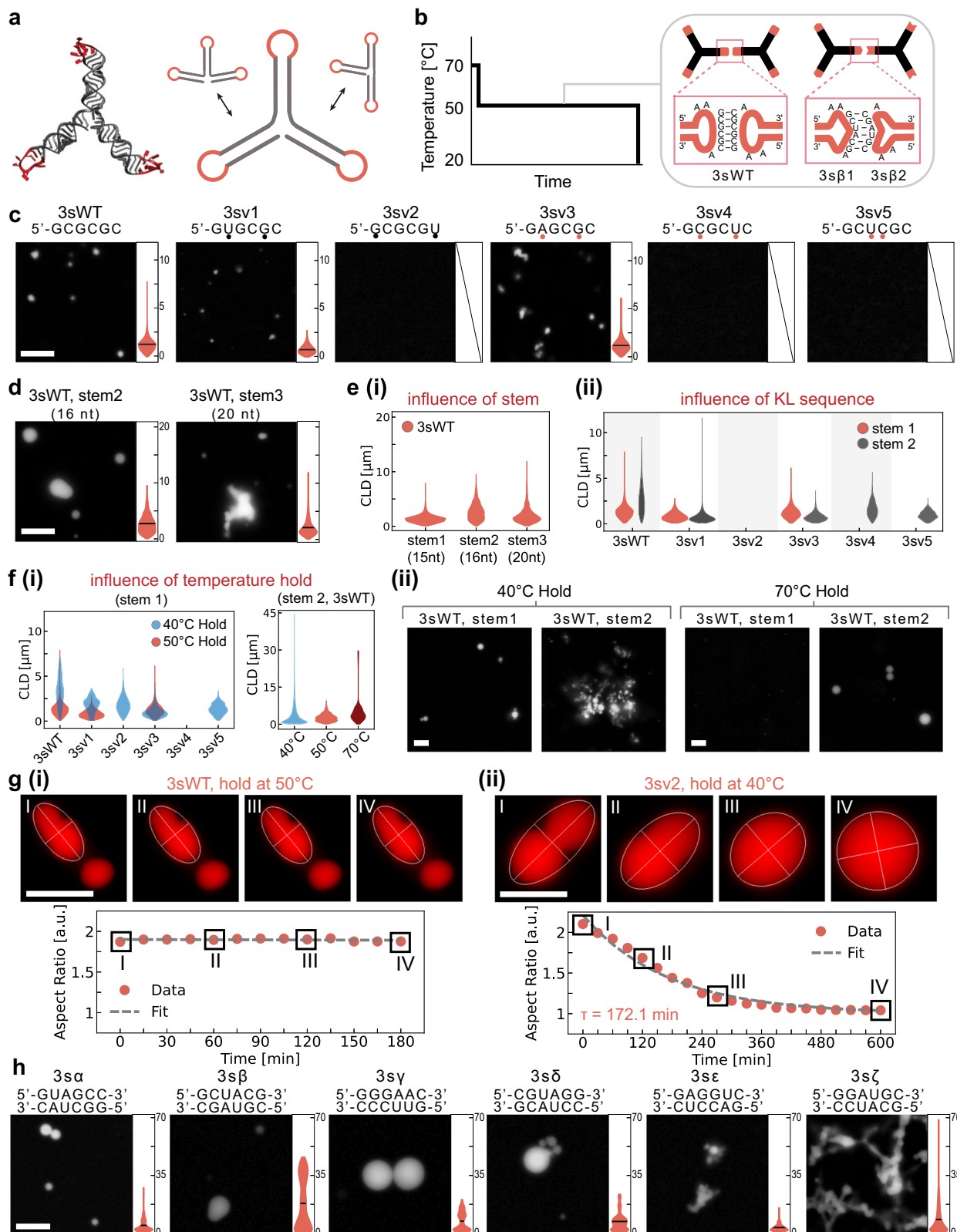

We found that condensate formation occurs robustly across designs, although it is influenced by KL and stem sequences. The 3sWT nanostars yield abundant condensates that continue to grow throughout the hold step of the annealing protocol (Supplementary Fig. 10). Condensate formation can be disrupted by replacing one KL with a polyadenine domain (Supplementary Fig. 11): this indicates that KL interactions, rather than stem-stem interactions, are the primary drivers for phase separation. We then tested 5 variants of the WT KL (dubbed 3sv1-v5) with mismatches or wobble pairs that modulate the probability of KL dimerization and therefore change the nanostar affinity (Fig. 3c and Supplementary Fig. 12). The selection and naming order of these variants is based on the dimerization capacity of the KL sequences taken from the HIV-1 dimer linker structure, as reported in literature[36]. Condensates are observed with the WT, v1, and v3 variants,

**Fig. 3 | Condensates generated by single-stranded RNA nanostars. a** PDB and 2D representation of a single-stranded RNA nanostar. **b** Melt and hold thermal annealing protocol; one-nanostar systems with palindromic KL (left) and two-nanostar systems with non-palindromic KL (right) condense during the hold phase. **c** Example images and violin plots of the chord length distribution (CLD) of condensates formed by nanostars differing by KL. Black dots indicate wobble pairs. Orange dots indicate mismatches. **d** Example images and CLD of condensates forming under different nanostar stems. **e** Influence of the stem sequence and length: (i) longer stems yield larger condensates; (ii) changing stem sequence impacts condensate formation and morphology across KL variants (**f**) (i) The hold temperature influences formation of stem1 variants; (ii) example images comparing 3sWT-stem1 and 3sWT-stem2 under a 40° and 70 °C hold. **g** Top: epifluorescence micrographs demonstrating coalescence of droplets from variant 3sWT-stem 1, held at 50 °C (i) and variant 3sv2-stem 1, held at 40 °C. Bottom: Time-dependent change in aspect ratios of condensates (orange dots), computed as the major and minor axes ratio from the best-fit-ellipses. Dashed lines are exponential fits of function $1 + Ae^{(-t/\tau)}$. **h** Two-nanostar systems produce condensates of varying morphology, as captured by violin plots of the CLD. All experiments were replicated at least three times; images are representative examples. Violin plots pool data from three independent replicates. Scale bars: 10 μm.

yielding similar CLDs, but not with variants v2, v4, and v5: these outcomes are consistent with previous dimerization studies[36]. While KL v2 is expected to dimerize, it did not yield condensates likely due to the high hold temperature and the lack of divalent cations in our assembly buffer. Denaturing PAGE of purified RNA samples shows the presence of expected transcription products, as well as truncated products of consistent length across variants (Supplementary Fig. 13). These truncated products appear to hinder condensation, as they increase the critical concentration for droplet formation when compared to gel extracted RNA samples that exclude the truncated products (Supplementary Fig. 14). Native PAGE of annealed samples show retention in the wells, confirming the presence of large, stable aggregates (Supplementary Fig. 15).

Next, we examined the influence of stem sequence and length on condensate formation. We built nanostars with a 16-nt "stem 2" variant whose sequence is adapted from a well-known design for three-arm DNA nanostars[17], and a 20-nt "stem 3" variant designed using NUPACK. We obtain condensates with both stem variants with the WT KL (Fig. 3d), and longer arms correlate with larger condensate size, an effect also observed with DNA nanostars[37] (Fig. 3e (i)). When combining stem 2 with different KL variants, we find that variants v4 and v5 yield condensates (Fig. 3e (ii)) pointing to the fact that the stem-mediated interactions may play a role in condensation under the melt and hold protocol. This role cannot be simply explained thermodynamically, given the similar ΔG of stem 1 and stem 2 formation (NUPACK estimates ΔG = −67.45 kcal/mol for stem 1 and ΔG = −68.53 kcal/mol for stem 2 at 50 °C, with comparable GC content flanking the KL). The stem sequences flanking the KL and flanking the NS core may create kinetic effects as they influence bond mobility, and could participate in domain swapping[38]. Interestingly, we also found that variant 3sv2 yields condensate when combined with longer stem 3 (Supplementary Fig. 12), while it does not with shorter stems 1 and 2 (Fig. 3e(ii)). Our results indicate that the stem length and sequence have major effects on the propensity of nanostars to form condensates: a model capturing these coupled effects could be obtained through further systematic experiments combined with molecular dynamics simulations[39].

The choice of hold temperature significantly influences condensation (Fig. 3f). For stem 1 variants, a 40 °C hold facilitates condensation when compared to a 50 °C hold: as shown in Fig. 3f(i), variants 3sv2 and 3sv5 form condensates at 40 °C but do not at 50 °C, and variants 3sWT and 3sv1 form larger droplets at 40 °C when compared to 50 °C (example microscopy images are in Supplementary Fig. 12). All stem 2 variants, except 3sv2, yield condensates under a 50 °C hold (Fig. 3e(ii) and Supplementary Fig. 12). Strikingly, variant 3sWT-stem2 forms condensates even with a 70 °C hold, likely due to interactions enabled by partial stem melting (Fig. 3f(i) and (ii)).

To quantify the mobility of nanostars in the dense phase, we performed Fluorescence Recovery After Photobleaching (FRAP) experiments for variants 3sWT at 50 °C and 3sv2 at 40 °C (Supplementary Fig. 16). Nanostars were fluorescent-tagged by doping 1% of CY3-UTP during transcription. No significant recovery was observed over five minutes. To study the kinetics of condensate fusion, we estimated the inverse capillary velocity $\eta/\gamma$, where $\eta$ is the viscosity

and $\gamma$ is the surface tension (Fig. 3g and Supplementary Fig. 17). Variant 3sWT showed no fusion throughout the 3-h incubation at 50 °C (Fig. 3g, left). Variant 3sv2 slowly fused over 10 hours and the example condensate shows a relaxation constant $\tau = 172$ minutes. We find $\eta/\gamma = 57.81$ min μm$^{-1}$ (Supplementary Fig. 17), which is significantly larger when compared to natural and artificial biomolecular condensates[40–43] as well as artificial DNA condensates[44]. Collectively, this evidence suggests that single-stranded nanostar condensates are viscous, like multi-stranded nanostars. Finally, lateral confocal projections showed no significant wetting of condensates on the glass slide surface (Supplementary Fig. 18).

The addition of MgCl$_2$ to the assembly buffer alters the observed phase transitions. In this case, also 3sWT-stem1 nanostars yield condensates under a 70 °C hold, and variant 3sv2-stem1 also produces condensates under a 50 °C hold (Supplementary Fig. 19). Other variants produce more non-spherical condensates when compared with the standard assembly buffer including monovalent cations (Supplementary Fig. 19). This behavior is consistent with the fact that divalent cations like MgCl$_2$ generally stabilize nucleic acid assemblies[45], and may increase the affinity of KL nanostars that otherwise do not condense[46]. At the same time, MgCl$_2$ can promote aggregation and kinetic trapping[47], altering condensate viscosity. This is an important consideration because MgCl$_2$ is typically required for in vitro transcription, and is a ubiquitous component in protocols for DNA or RNA self-assembly.

Finally, we tested 6 more RNA nanostar variants (3sα-ζ) each comprising two distinct nanostars (e.g., 3sα1, 3sα2) with non-palindromic KL designed to be complementary (Fig. 3h). By adopting non-palindromic sequences, we can expand by 64-fold the theoretical sequence design space of a 6 nucleotide KL domain. With the melt and hold protocol and our assembly buffer, we found that all two-nanostar variants generated condensates with variable size and morphology (Fig. 3h). While individual nanostars (e.g., 3sα1) have the potential to multimerize as they carry identical stems, they fail to produce any condensate except for variant 3sε1, which we attribute to partial self-complementarity of the KL domains (Supplementary Fig. 20). Native PAGE shows that two-nanostar samples are retained in the well, consistent with the formation of condensates. When only one nanostar out of the two is present, the sample migrates in the well and generates a few bands likely to be multimers forming due to stem-stem interactions (Supplementary Fig. 15). These findings indicate that stem-mediated RNA multimerization can occur but is not the primary driver of condensation, unlike what proposed in the recent literature[6,48].

In summary, our data shows that changes in KL and stem length and sequences all influence condensation. KL with more than 4 base pairs is likely to yield condensation, depending on the overall strength of their interaction (Supplementary Fig. 21), which is influenced by non-canonical base pairing, stacking, and secondary and tertiary structure[6].

## Orthogonal RNA condensates can be programmed to recruit guest molecules

We next demonstrate the potential of RNA nanostars to capture client molecules and recruit them to the RNA-dense phase. We focus on

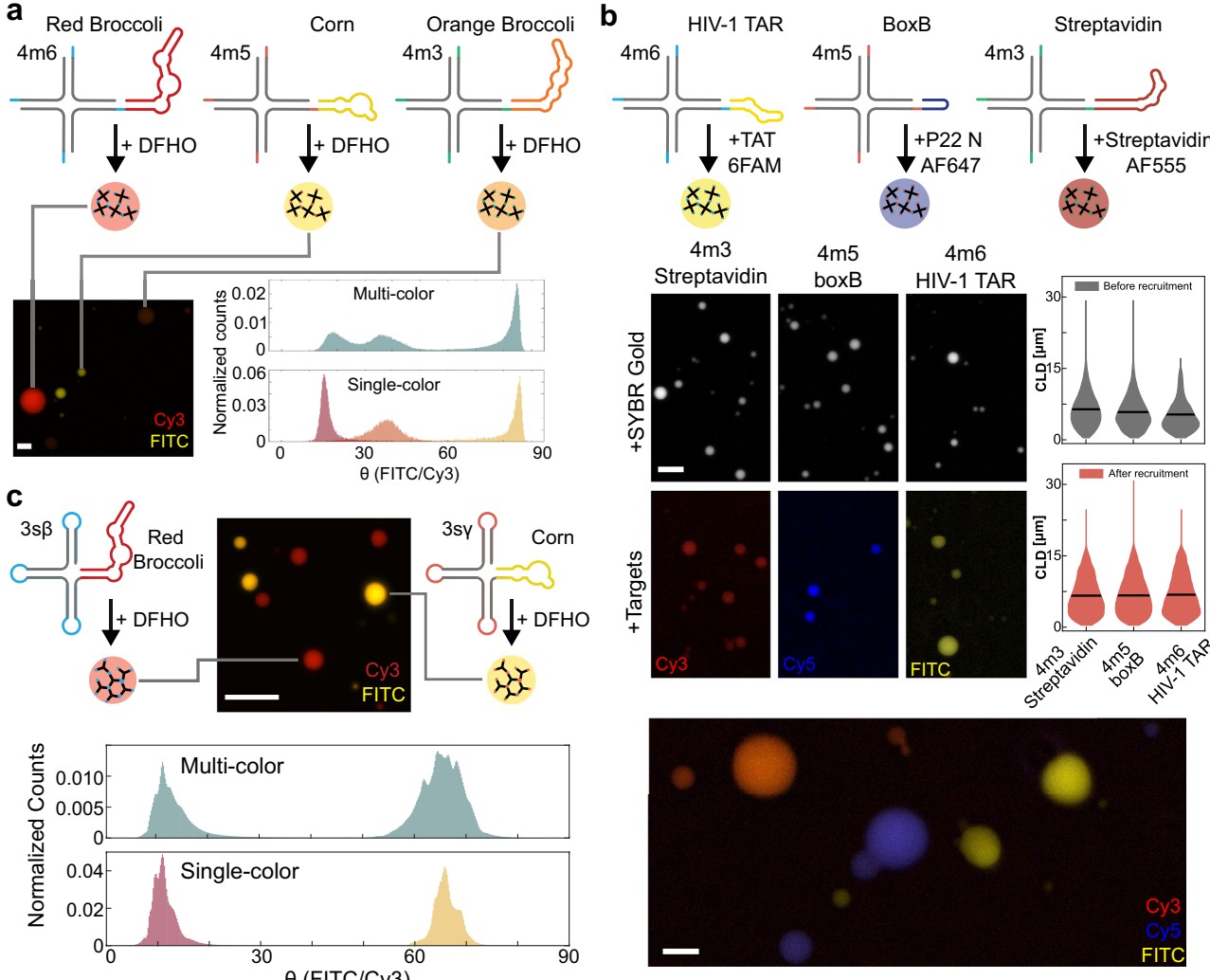

**Fig. 4 | Sequence-orthogonal RNA nanostars produce distinct condensates that recruit clients specifically and remain demixed. a** Multi-stranded sequence-orthogonal nanostars modified to include FLAPs. When annealed in one pot they produce distinct RNA condensates that do not mix. Condensates were imaged in both FITC and Cy3 channels; each pixel's FITC/Cy3 ratio was mapped to an angle θ and histogrammed. Histogram peaks for nanostars annealed individually align with those of nanostars annealed together. **b** Top: Nanostars including aptamer domains that recruit streptavidin, P22 N peptide, and TAT peptide guests. Middle: Example images and violin plots of the CLD for condensates before and after recruitment of peptides. Black line indicates the mean. Bottom: Nanostars were designed to be sequence-orthogonal, thus their one-pot assembly with fluorescently labeled clients shows their recruitment and compartmentalized into specific condensates (red, blue, and yellow). **c** Sequence-orthogonal single-stranded RNA motifs pairs (two nanostar systems, 3sβ and 3sγ), modified to include Corn and Red Broccoli fluorogenic aptamers. The peaks of the FITC/Cy3 angle histogram for nanostars annealed individually align with the peaks of nanostars annealed together. Experiments were done in triplicates; images provide representative examples. Violin plots and histograms pool data from all replicates. Scale bars: 20 μm.

condensates produced through the melt-and-hold protocol using purified RNA.

A major advantage of multi-stranded nanostars is that only one of the strands needs to be modified for the inclusion of aptamers, so we began by appending client recruitment domains (aptamers) at the tip of one of the arms of variants 4m3, 4m5, and 4m6 (upstream of the 5' end of the sticky-end). We first validated this idea through fluorescent light-up aptamers (FLAPs) known as Red Broccoli (appended to variant 4m6, Corn (4m5), and Orange Broccoli (4m3), which all bind to fluorophore 3,5-difluoro-4-hydroxybenzylidene imidazolinone-2-oxime (DFHO) but each results in a distinct emission spectrum (Fig. 4a, Supplementary Fig. 8). (For ease of visualization, we depict FLAPs using the secondary structure predicted by NUPACK[26], which does not capture their complex 3D tertiary structure[49]). Because these variants were designed to maximize orthogonality, we expected each nanostar to form distinct condensates that do not mix with the others. Through microscopy, we first verified that these nanostars yield

condensates when assembled in isolation in the presence of DFHO (Supplementary Fig. 8), and then that they do not mix when assembled simultaneously. To quantitatively assess the degree of condensate mixing we plotted a histogram of the arctangent of the pixel intensity ratio FITC/Cy3 (Fig. 4a, bottom right, and Supplementary Fig. 22). The histogram peaks of control samples (individually assembled nanostars) are consistent with those of samples including mixed nanostars, indicating that the aptamers are not colocalized. These histograms highlight that although our nanostars are designed to remain demixed, the excitation and emission spectra of the FLAPs have some overlap (Orange Broccoli 513/562 nm, Red Broccoli 518/582 nm, and Corn 505/545 nm)[49]. Further, Orange Broccoli and Red Broccoli have a high degree of sequence similarity, where mutation at nucleotide position 71 is hypothesized to be the key cause of the difference in fluorescence emission[49,50]. Finally, differences in fluorescence intensity are likely due to the difference in the $K_D$ of DFHO (Orange Broccoli ~230 nM, Red Broccoli ~206 nM, and Corn ~70 nM[49,50]).

Next, we show that our RNA condensates can recruit client peptides to the dense phase. For this purpose, we replaced FLAPs with peptide and protein binding domains: we considered a streptavidin binding aptamer (appended to nanostar variant 4m3), boxB RNA that binds P22 N peptide (appended to nanostar variant 4m5), and TAR RNA that binds TAT peptide (appended to nanostar variant 4m6)[51–53], shown in Fig. 4B. We selected these peptides for their versatility: the streptavidin-biotin pair is widely used for purification and localization assays[54]; the P22 bacteriophage N peptide is a strong binder for its aptamer boxB, and is useful for RNA colocalization studies[55]; finally, Tat bound to its aptamer is a strong transcriptional regulator involved in HIV replication[56]. Strands including aptamers were doped into the mixture of unmodified strands at a ratio of 1:4 (25%). We verified that each peptide binding nanostar yields condensates when assembled individually, and it successfully recruits its client; after peptide recruitment, condensates become on average larger (Fig. 4b, bottom). Undesired cross-binding of targets to their non-cognate aptamer occurs only for TAT, due to its positively charged amino acid residues that cause non-specific binding to negatively charged RNA (Supplementary Fig. 9). Overall, our orthogonal RNA nanostars partition the fluorescently labeled peptides into distinct compartments that do not mix.

Finally, we confirmed that also single-stranded RNA nanostars can recruit client molecules. We selected orthogonal two-nanostar variants 3sβ and 3sγ and we modified them to include FLAPs binding DFHO (Red Broccoli and Corn, respectively) as an additional arm (25% FLAP-modified strands in the mix). We observed the formation of condensates of distinct colors upon annealing nanostars together (Fig. 4c, Supplementary Fig. 23). We verified that these condensates do not mix through our quantitative image analysis (computation of the arctangent of FITC/Cy3 intensity ratio for each pixel, Supplementary Fig. 22), which shows aligned histogram peaks when purified nanostars are annealed separately or together (Fig.4c, bottom).

### Co-transcriptional formation of RNA condensates

We finally demonstrate that single-stranded RNA nanostars produce condensates co-transcriptionally at 37 °C, in the absence of thermal annealing (Fig. 5a). Tandem stem-loops fold based on local RNA interactions so they are expected to form stable structures as they are being transcribed under isothermal conditions[57]. We transcribed nanostars using linear templates under the control of the T7 bacteriophage promoter, using a high-yield in vitro transcription buffer that includes components required for transcription and 10 mM NaCl, 30 mM MgCl$_2$, and 2 mM spermidine[58] (see "Methods"). For these experiments, we selected variant 3sv2-stem1 (Fig. 5a), which produces abundant spherical condensates when purified and thermally treated with a 40 °C hold, as well as in the presence of MgCl$_2$ (Supplementary Figs. 12 and 19). We observed large condensates (stained with SYBR Gold) within 1–2 hours of nanostar transcription (Fig. 5b), which corresponds to an estimated RNA concentration of 2 μM (Supplementary Fig. 24). The amount of RNA produced correlates with average condensate size, measured through the mean-chord length (μ$_{CLD}$), and both double within the first 3 hours; larger aggregates can be obtained by increasing the DNA template concentration (Supplementary Fig. 25). The number of condensates only slightly decreases over time, which suggests that our reaction conditions promote condensate growth primarily by monomer addition, rather than fusion, while nanostars are being transcribed (Fig. 5b). FRAP of condensate produced during transcription shows no recovery over the observation window (Supplementary Fig. 26). We also tested the co-transcriptional assembly of the two-nanostar variant 3sβ, which similarly produces aggregates that grow rapidly into a slowly fusing network (Fig. 5c, d); 3sβ1 or 3sβ2 transcribed individually do not form condensates (Supplementary Fig. 27). Native PAGE confirms the formation of large aggregates in the two-nanostar sample, and some multimerization for

3sβ1 or 3sβ2, as observed on purified and annealed samples (Supplementary Fig. 28). We found that the amount of monovalent and divalent cations (NaCl and MgCl$_2$) in the transcription buffer has significant effects on co-transcriptional formation, as shown in control experiments involving the two-nanostar variant 3sβ (Supplementary Fig. 29); buffers included in commercial kits may fail to yield condensates if they do not include sufficient cation levels. We expect that co-transcriptional condensates can be obtained using other nanostar variants that produce condensates from purified RNA with the melt and 40 °C hold protocol (Fig. 3f, Supplementary Fig. 12).

co-transcriptionally produced RNA condensates could be useful as membrane-less organelles that can spontaneously recruit proteins. To demonstrate that this is immediately feasible in cell-free systems, we verified that nanostar 3sv2-stem1 could produce condensates when transcribed using PURExpress®, a commercial kit for cell-free protein synthesis[59]. The production of large condensates appears slow in PURExpress® when compared to a high-yield transcription kit, as evidenced by example images of Fig. 5d and by the μ$_{CLD}$ shown in Fig. 5f. Condensate growth rate may be increased by using larger amounts of DNA template or of RNA polymerase. After several hours, PURExpress® samples form very large, spherical condensates when compared to high-yield in vitro transcription conditions; this difference may be due to the presence of translation components of the PURExpress® kit (Supplementary Fig. 30). Finally, we modified 3sv2-stem1 to recruit p22 N and TAT peptides using their corresponding aptamers (boxB and TAR, respectively) and demonstrated that these peptides are recruited to the condensates co-transcriptionally. We incubated peptides and DNA templates for 2 h in high-yield transcription buffer, and we verified the correct, specific recruitment of peptides to the dense phase (Fig. 5g). No qualitative difference is observed if peptides are added 2 h after the start of transcription (Supplementary Fig. 31).

## Discussion

We have demonstrated the design and synthesis of modular RNA nanostars for phase separation. We showed that both multi-stranded and single-stranded nanostars robustly form condensates in standard buffers commonly adopted in nanotechnology applications, in the absence of binding partners. We focused on two protocols, (1) transcribing and purifying RNA strands, then performing a temperature treatment consisting of a denaturing step and a long temperature hold, or (2) co-transcriptional assembly. We examined how condensation is influenced by various nanostar design features and extrinsic factors such as ionic conditions and temperature. Further, we have programmed orthogonal condensates to recruit and organize small molecules and peptides, thus mimicking the ability of biological condensates to recruit clients[60]. Our strategy is modular, and may lead to the development of libraries of orthogonal condensates recruiting diverse clients. Multi-stranded and single-stranded RNA nanostars offer different advantages depending on the downstream application or purpose. Owing to the ease in designing linear sticky-ends and hybridizing domains, the affinity and valency of multi-stranded nanostars can be easily tuned, making it possible to build RNA condensates with a broad range of biophysical properties that should be comparable to those demonstrated for similar DNA condensates[19,20,27]. Further, this design allows for the modular introduction of RNA or DNA strands with distinct functionalities, including client recruitment and adaptation to chemical or physical inputs that are relevant for developing therapeutic, multifunctional biomaterials[61]. Single-stranded nanostars produce condensates isothermally under physiological conditions, making them immediately useful as RNA organelles that can be produced in artificial cells, as demonstrated in a parallel study[25]. We are evaluating whether these constructs can be genetically encoded in living cells.

A distinguishing feature of our RNA condensates is that they are formed through compact, nanostructured motifs designed using short

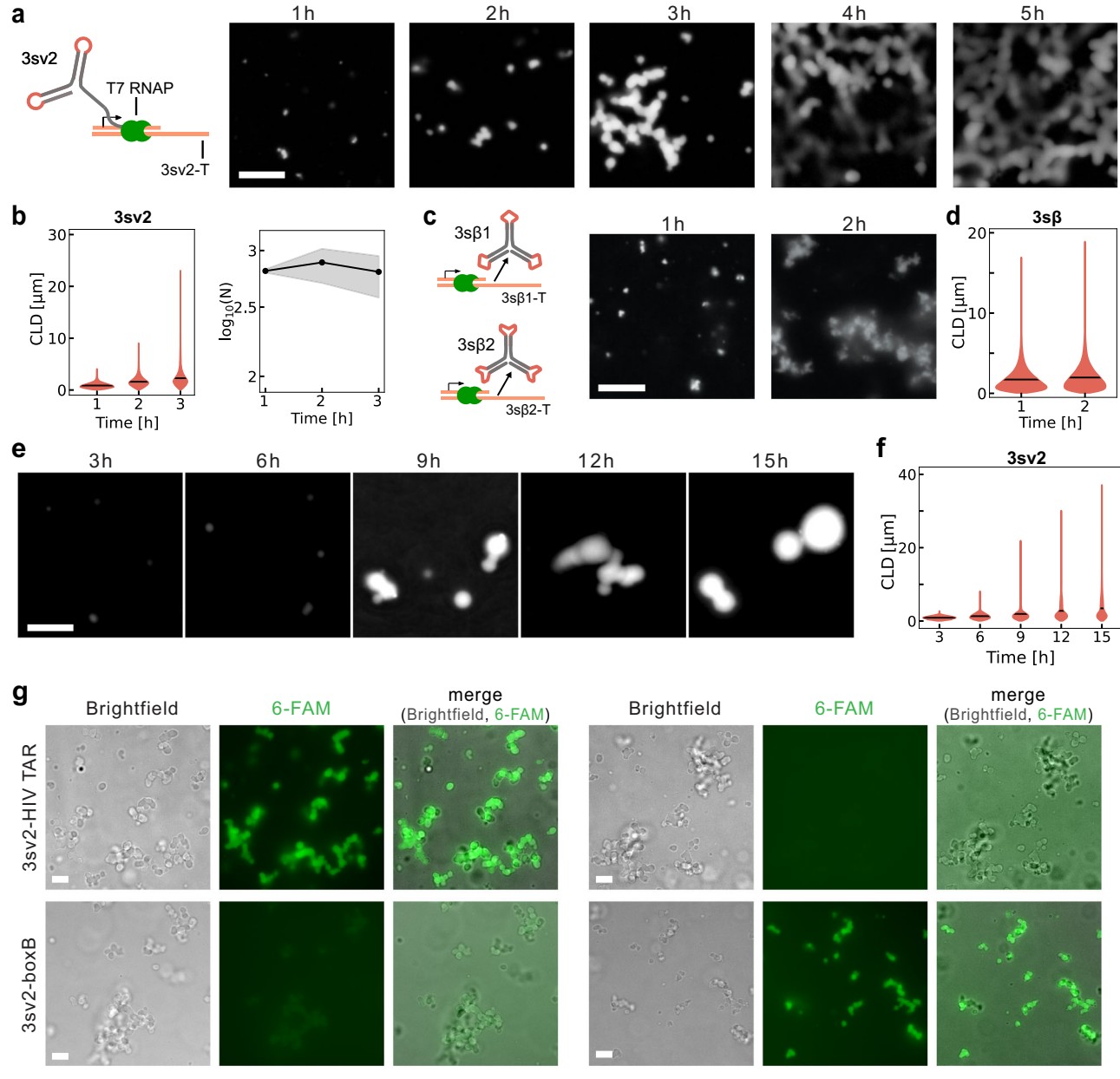

**Fig. 5 | co-transcriptional, isothermal formation of single-stranded RNA nanostars and peptide recruitment. a** Scheme shows transcription and co-transcriptional folding of motif 3sv2; example images show condensate formation in T7 in vitro transcription reaction sampled at different time points. **b** Temporal evolution of the CLD (left) and number (N, right) of 3sv2 nanostars. Black lines indicate the mean chord length from $n = 3$ at given time points. **c** Scheme shows transcription and co-transcriptional folding of motif pairs 3sβ1 and 3sβ2; example images show 3sβ condensate growth in T7 in vitro transcription reaction. **d** Temporal evolution of the CLD of 3sβ condensates. Black lines indicate the mean chord length from $n = 3$ at given time points. (**e**) Example images show 3sv2 condensate growth in a cell-free PURExpress® reaction. **f** Temporal evolution of the CLD of 3sv2 condensate in PURExpress®. Black lines indicate the mean chord length from $n = 3$ at given time points. **g** Brightfield and fluorescence microscopy images show specific peptide recruitment to condensates formed co-transcriptionally after 2-hour incubation. 3sv2 nanostars were modified to include TAR (top) or boxB (bottom) aptamer, recruiting either TAT or p22 N peptide labeled with 6-FAM. Peptides are added at the beginning of transcription. Experiments were done in triplicates; images are representative examples. Scale bars: 10 μm.

sequences that are optimized to fold as desired. Previous work has shown the emergence of coacervates from unstructured RNA homopolymers in the presence of polymeric cations that are not necessary in our system[62–64]. RNA strands assembling into pure RNA hydrogels have been identified via SELEX[65], however, this approach does not offer a clear design strategy to modularly adapt the condensing motifs as easily as with RNA nanostars. Recent work demonstrated that liquid RNA droplets can be produced inside cells through long RNA molecules featuring short sequence repeats (CAG/CUG)[15,16,66]. By fusing the CAG-repeats to recruitment domains, it was possible to demonstrate

compositional control of RNA droplets both in bacteria and in mammalian cells[15,16]. Intracellular RNA condensates were also obtained via homodimerization of two different RNA aptamer repeats[67], which remain demixed forming up to two types of orthogonal compartments inside cells. In these exciting achievements, sequence repeats are an expedient strategy to introduce multivalency, but it is not clear this approach can yield a scalable number of orthogonal condensates.

A simplified working model of RNA nanostars suggests that we can determine their affinity, valency, and size by changing the sticky ends/kissing loops, number of arms, and arm length. However, we still

lack a clear picture of how different combinations of these parameters affect phase transitions and condensation kinetics. Further, some of these parameters mutually couple in ways that are difficult to predict; for example, changes in the stem sequence can have an impact on the nanostar affinity if they affect the folding of the interaction domains, and these effects are sensitive to temperature and ionic conditions. The influence of combinations of design changes could be predicted by coarse-grained models that capture the interactions among nanostars[68,69], and should be validated against systematic experiments characterizing phase diagrams and growth rates of nanostar condensates. These models could accelerate the design of customized RNA condensates while minimizing experimental burden. Finally, while we have demonstrated modular recruitment of small molecules and peptides, we expect limitations in which classes of molecules of proteins are recruited by RNA nanostars with specificity. RNA aptamers can be promiscuous, for example TAR RNA is known to bind positively charged arginine[70]. Further, arbitrary RNA domains may non-specifically recruit amino acid sequences; for example, positively charged peptides naturally tend to bind to the negatively charged backbone of RNA, regardless of sequence specificity (like we observed with TAT protein). We expect that, in complex biological samples, various RNA-binding proteins could bind non-specifically to our artificial condensates and possibly compromise their formation. Another challenge is that some recruitment motifs could introduce misfolding or remodeling of the secondary structure of co-transcriptionally produced single-stranded nanostars. Design methods developed for RNA origami may provide valuable lessons toward enhancing the robustness and functionalities of RNA nanostars[71].

Orthogonal, customizable micron-sized compartments made with RNA may find clinical and industrial applications for purification, diagnostics, and therapeutic treatments. RNA condensates with the capacity to selectively sequester binding targets may be useful for separating crude mixtures of molecules or low-volume clinical samples. The incorporation of stimuli-responsive domains in RNA nanostars could make it possible to develop sensors and diagnostic tools, and further engineering may allow this platform to serve as a smart drug delivery system capable of encapsulating and releasing therapeutic molecules. Lastly, we envision that single-stranded RNA nanostars may be used to produce functional RNA organelles in living cells and potentially control cellular processes.

## Methods

Sequences, detailed methods, and additional experiments are included in the Supplementary Table and Supplementary Information file.

### Sequence design

The RNA strands were designed and optimized using the NUPACK design tool using scripts reported in Supplementary Note 1[26]. Aptamer sequences were selected from previous studies[49,50]. The Corn aptamer was modified from previous work 3 by adding GG to the 5′ end and CC to the 3′ end to ensure transcription. Red Broccoli and Orange Broccoli aptamers were modified from 2 to include flanking sequences GUAUGUGG at the 5′ end and CCCACAUAC at the 3′ end, which surround conserved fluorogen-binding sequences and broccoli bridging sequences. The boxB RNA sequence that binds P22 N peptide and HIV-1 Tat aptamer that binds Tat peptide was taken from previous work[72,73]. The streptavidin binding aptamer was modified from previous work[51] by adding a G at the 5′ end and a C at the 3′ end to ensure transcription. Aptamer sequences were appended to the 5′- sticky end of the S1 strand of a given motif. The secondary structure of the motif with aptamer modification was analyzed using NUPACK to ensure that the secondary structure of the motif and aptamer was similar to the fold of the individual modules. Three-armed single-stranded RNA nanostars (3 s) were designed by combining distinct arm sequences with several kissing loop variants. Stem 1 sequences were adapted from the multi-

stranded motifs (4 m); stem 2 sequences were adapted from the design by Sato et al. for DNA nanostars[17]; stem 3 sequences were designed using NUPACK. Two of the three spacers between arms include two unpaired adenines, and the other is a nick, allowing the motif to have flexible configurations. All KLs are nine nucleotides long and include a six-nt interaction sequence flanked by three unpaired adenine residues, two upstream and one downstream of the interaction sequence (5′-AA...A-3′). KL variants used for the one-nanostar motifs were adapted from the HIV-1 palindromic KL sequence[36]. Each variant was obtained by introducing a single-base substitution in the six nt interaction domain of the wild type KL (5′-GCGCGC). KL for two-nanostar motifs were designed de novo to function as pairs (heterodimers). All sequences for the two-nanostar design include four GC pairs and two AU pairs to ensure similar bond stabilities. Fluorogenic aptamer sequences identical to the multi-stranded designs were adopted from literature[49,50]. The expected folding of each design was confirmed using NUPACK[26]. All sequences are listed in Supplementary Data 1.

### RNA synthesis

RNA strands for multi-stranded nanostars were transcribed from PAGE purified DNA templates, including a T7 promoter, purchased from Integrated DNA Technologies. Lyophilized DNA was resuspended in nuclease-free water. RNA strands for single-stranded nanostars were transcribed from standard desalt DNA templates, purchased from Integrated DNA Technologies as Lab Ready. DNA templates were annealed in 1X TE/50 mM NaCl from 90 °C to RT at −1 °C/min. RNA strands were individually transcribed in vitro using the AmpliScribe T7-Flash transcription (ASF3507, Biosearch Technologies) kit from DNA templates following the manufacturer's protocol. RNA strands were then purified using Amicon Ultra 10 K 0.5 ml centrifugal filters (UFC501096) and 1X TE buffer and centrifuging three times at 14,000 g. For multi-stranded nanostars, each strand was transcribed from a fully double-stranded DNA template, including the T7 promoter. For single-stranded nanostars, we used single-stranded non-coding DNA templates annealed to a 21-nt complement including the promoter region and a 4 nt sealing domain (5′-GCGC). We estimated the concentration of purified RNA using Nanodrop 2000c from measurements of absorption at 260 nm and the extinction coefficient provided by the manufacturer.

### Condensate preparation

Condensates made from purified RNA (multi-stranded or single-stranded nanostars) were formed in our assembly buffer: 40 mM HEPES, 100 mM KCl, 500 mM NaCl. Strands were thermally annealed in an Eppendorf Mastercycler using a melt and hold protocol which includes a melt at 70 °C for 10 minutes, followed by 12 hours of incubation at specified temperatures, and by a quick drop to 20 °C for 5 minutes before imaging. For multi-stranded nanostars, we used purified RNA strands each at equimolar concentrations (5 μM). Strands including aptamer sequences were added at 1.25 μM concentration (25% doping). For single-stranded nanostars, we used purified RNA at an estimated 5 μM concentration. For two-nanostar condensates we used purified RNA with each nanostar at 5 μM concentration; thus, the total RNA concentration in these experiments was doubled. For two-nanostar motif experiments using fluorogenic RNA aptamers, aptamers-containing strands were added at a 25% doping. Condensates formed from motifs without aptamer were imaged with 1X SYBR Golding staining (S11494, Thermofisher). Dye was added 5 minutes after cooling to 20 °C. Condensates formed from aptamer-appended motifs were stained with DFHO (Lucerna Technologies). DFHO stock was stored in DMSO at a concentration of 10 mM. DFHO staining solution was prepared by diluting DFHO stock in HEPES buffer with a final concentration of 1 mM DFHO and 40 mM HEPES. Samples were imaged immediately after staining.

## Preparation of peptide and protein targets

Fluorescent peptides were synthesized by either LifeTein, LLC or GenScript. Fluorophores were added to the N-terminal, AlexaFluor647 for P22 N peptide, and 6-FAM for TAT peptide. For co-transcriptional experiments, both N peptide and TAT were labeled with 6-FAM. Peptide synthesis was guaranteed a purity of ≥95% with standard Trifluoroacetic acid (TFA) removal (Final TFA Counterion % <10%. Lyophilized peptide was resuspended in 10 mM HEPES with a molarity of ~350 μM and stored at 5 °C. Streptavidin, Alexa Fluor™ 555 conjugate was purchased from Thermo Fisher Scientific at 2 mg/ml concentration, with a molarity of ~35 μM. Streptavidin conjugate was diluted with 10 mM HEPES and stored at 5 °C. For multi-stranded nanostars, peptides were added to the samples after the melt-and-hold step. For single-stranded nanostars, peptides were added at the beginning of co-transcription.

## Co-transcriptional production of condensates

Condensates produced during transcription of single-stranded RNA nanostars were formed using DNA templates prepared as described under "RNA synthesis". RNA strands were transcribed in vitro at 37 °C using 7.5% (v/v) T7 polymerase from the AmpliScribe T7-Flash transcription kit (ASF3507, Biosearch Technologies), and transcription buffer prepared in-house: 40 mM of Tris-HCl, 10 mM of NaCl, 30 mM MgCl$_2$, 2 mM spermidine, 7.5 mM each NTP, 10 mM DTT. For the sequence-orthogonal single-stranded RNA nanostars (two-nanostar system), the RNA strands were transcribed in vitro under the same conditions, with the exception of the NaCl and MgCl$_2$ concentrations, which were adjusted to 20 mM each. When using the PURExpress® kit (E6800S, New England Biolabs), we adhered to the manufacturer's protocol and incubated our sample at 37 °C. Unless otherwise specified, we used a final concentration of 10 nM DNA template for the in vitro transcription; we used 10 ng DNA for the cell-free PURExpress® reaction.

## Fluorescence microscopy

Condensates produced from multi-stranded motifs were obtained with an Olympus BX-UCB upright fluorescence microscope using a 20x air objective. We used filtersets Cy3 (Chroma Filter Set Exciter D540/25x EX Dichroic Q565lp BS Emitter D620/60 m EM), Cy5 (Chroma Filter Set Exciter HQ620/60x EX Dichroic Q660LP BS Emitter HQ700/75 m EM), and FITC (Chroma Filter Set Exciter D480/30x EX Dichroic Q505lp BS Emitter D535/40 m EM), with a standard exposure time of 100 ms for samples stained with SYBR gold, 500 ms for samples with DFHO or fluorescent peptides and proteins. Condensates produced from single-stranded motifs were imaged using a Nikon Eclipse TI-E inverted microscope using a 60x oil immersion objective. SYBR gold-stained samples were detected using the FITC channel (ex 455−485 nm/em 510−545 nm) with an exposure time of 100 ms. DFHO-stained samples were detected in both the FITC channel with an exposure time of 200 ms, and in the Cy3 channel (ex 512−552 nm/em 565−615 nm) with an exposure time of 100 ms. Peptide recruiting condensates, which contain 6-FAM labeled peptides, were imaged using the FITC channel (ex 455−485 nm/em 510−545 nm); exposure time was set to 500 ms, except for the samples including 3sv2-TAT nanostars and TAT peptide, in which exposure was set at 100 ms. For samples stained with SYBR Gold, 1X SYBR gold was used after thermal treatment. For samples with DFHO, 50 μM DFHO in 2 mM HEPES buffer was used before thermal treatment (for multi-stranded nanostars) and after thermal treatment (for single-stranded nanostars). For lateral confocal projections, samples were annealed using the melt-and-hold protocol and stained with 1x SYBR Gold, and imaged under a 60x objective on a Nikon Ti-2 microscope with an NLS5+ camera. Z-stack images were captured with a thickness of 0.3 μm.

## Image processing and visualization

All fluorescence images were pre-processed in FIJI (ImageJ). All codes are provided as indicated in the code availability. Raw images were background subtracted, contrast-enhanced, and converted to a binary mask. For condensate number analysis, objects smaller than 6 px$^2$ were considered noise and excluded. Condensate numbers were measured by FIJI and recorded. To gather information on condensate size, we measured chord length distributions (CLD)[32–35] from the binary masks using a Python3 script based on PoreSpy, which relies on Scipy and Skimage. Violin plots were generated using a Python3 script based on Seaborn. All experimental replicates were pooled into a single violin plot. Means were computed across three technical replicates. To determine whether our nanostars including distinct fluorogenic aptamers produce condensates that mix or do not mix, we built pixel intensity histograms from fluorescence microscopy images (Fig. 4, Supplementary Fig. 22). This was done because upon DFHO staining, both Corn and Red Broccoli aptamers can be detected in the FITC channel and the Cy3 channel, albeit with varying intensity, since the Corn aptamer emission peak is 545 nm, and the Red Broccoli emission peak is 582 nm. Histograms were generated from the coordinate angles calculated from every pixel within the regions of interest. Additional details on image processing and visualization are provided in Supplementary Note 2.

## Denaturing Polyacrylamide Gel Electrophoresis

Gel pre-mix was prepared by adding 42 g of urea to nanopure water, the mixture was then heated until the urea completely dissolved. This mixture was allowed to cool to room temperature, and then a 40% (v/v) 19:1 acrylamide/bis-acrylamide solution was added in the appropriate volume for the desired percentage (final volume 100 mL). 8 mL of pre-mix was added in appropriate ratios with TBE and nanopure water, ammonium persulfate (APS), and tetramethylethylenediamine (TEMED) to start polymerization. Gels were cast in 8 × 8 cm, 1 mm thick disposable mini gel cassettes (Thermo Scientific, #NC2010) and allowed to polymerize for 30 minutes to 2 h before electrophoresis. Samples and low-range ssRNA ladder (NEB, N0364S) were prepared by mixing individual strands with denaturing RNA loading dye (NEB, B0363S), then heated at 90 °C for 10 minutes and immediately placed on ice. After curing, the gel was pre-run in a 1X TBE buffer for 30 minutes. Gels were run at room temperature at 100 V in 1X TBE unless otherwise noted. After electrophoresis, the gels were stained in 1xSYBR Gold Nucleic Acid Gel Stain and then imaged using the iBright™ FL1500 Imaging for multi-stranded nanostars and using the Bio-Rad Gel Imaging Systems for single-stranded nanostar.

## Non-denaturing Polyacrylamide Gel Electrophoresis

40% solution of 19:1 acrylamide/bis-acrylamide, TAE, APS, and TEMED were added together at appropriate concentrations for the desired polyacrylamide percentage, then cast in 8 × 8 cm, 1 mm thick disposable mini gel cassettes (Thermo Scientific, #NC2010) and allowed to polymerize for at least 2 h before electrophoresis. Samples were prepared by annealing in our assembly buffer (40 mM HEPES/100KCl/500 mM NaCl) in 1:1 stoichiometric ratios. Gels were run at 4 °C at 120 V in 1X TBE buffer. After electrophoresis gels were stained in SYBR Gold Nucleic Acid Gel Stain or ethidium bromide and then imaged using the iBright™ FL1500 Imaging for multi-stranded nanostars and using the Bio-Rad Gel Imaging Systems for single-stranded

## Fluorescence recovery after photobleaching (FRAP)

RNA was transcribed by adding 1% of CY3-labeled UTP (ENZ-42505) to the transcription mix. Co-transcribed samples were diluted 10 times with 1x transcription buffer after transcription to reduce background fluorescence. Samples were loaded into a house-made chamber and sealed with epoxy (Gorilla, 5-minute set) for imaging. Imaging was

done every 5 seconds for 30 seconds before bleaching and every 5 seconds for 5 minutes after bleaching. Bleaching was done with a 488 nm laser, at 50 ms exposure for multi-stranded nanostars and 200 ms exposure for single-stranded nanostars. Images were processed to eliminate the influence of horizontal drifting using the SIFT algorithm[74]. Normalized pixel intensity within ROIs was exported and recovery was calculated as

$$(I_{bleach,t}/I_{bleach,max})/(I_{unbleach,t}/I_{unbleach,max}) \qquad (1)$$

where $I$ denotes the mean pixel intensity among the bleached or unbleached area, $t$ denotes the time point, $max$ denotes the highest pixel intensity within the area among all time points. Images are plotted as mean ± error bar from N = 3.

### Condensate fusion
Samples were purified, annealed, stained with 1x SYBR Gold, then loaded into an observation chamber and sealed with epoxy as described above. Imaging was done every 15 minutes for 4 hours under the FITC channel. Each fusion event was identified manually and segmented using Otsu thresholding. The binary mask was then labeled to extract the centroid position, major and minor axis lengths, and orientation. Best-fit-ellipses were generated based on the extracted data. The aspect ratio was calculated as the major-to-minor axes ratio and used for further analysis. A detailed description of image analysis and Python packages used is in Supplementary Note 2, Data Processing for time-dependent coalescence.

### Statistics & Reproducibility
For peptide recruitment samples using multi-stranded nanostars, size analysis of each condition was performed using data from $n = 3$ and fields of view (FOV) = 10, for incubation over time samples of multi-stranded nanostars, size analysis of each time point was performed using data from one experimental replicate and FOV = 14. For single-stranded nanostars, $n = 3$ is applied unless specified. Each experiment with purified RNA includes FOV = 7; formation of orthogonal Corn/Red Broccoli-tagged condensates includes FOV = 10; each co-transcription experiment includes FOV = 11; each PurExpress co-transcription experiment includes FOV = 11. Images with very strong background noise, making data processing impossible are excluded. To ensure the same sample sizes, the final number of images for the experiment depends on the group with the minimum processable data.

### Reporting summary
Further information on research design is available in the Nature Portfolio Reporting Summary linked to this article.

## Data availability
Data required to generate the figures are provided either in the Source Data file or the GitHub repository: https://github.com/FrancoLabUCLA/Stewart-Li-Tang-2024-Nat-Commun. Due to file sizes, unprocessed microscopy images are available upon request from the corresponding authors, who will respond within one business week local time. Source data are provided in this paper.

## Code availability
Image analysis was done using publicly available packages as described in the Supplementary Methods. Custom code is available at: https://github.com/FrancoLabUCLA/Stewart-Li-Tang-2024-Nat-Commun.

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

## Acknowledgements

JMS is a Merck Awardee of the Life Sciences Research Foundation. EF acknowledges support from the Alfred Sloan Foundation through award G-2021-16831, and from the US NSF through CAREER award 1938194, FMRG: Bio award 2134772, and BBSRC-NSF/BIO award 2020039. EF also acknowledges funding from the UCLA Eli and Edythe Broad Center of Regenerative Medicine and Stem Cell Research Rose Hills Foundation Innovator Grant. PWKR acknowledges support from the Alfred Sloan Foundation through award G-2021-16831 and from the US NSF through FMRG: Bio award 2134772. LDM acknowledges support from the European Research Council (ERC) under the Horizon 2020 Research and Innovation Program (ERC-STG No 851667 – NANOCELL) and a Royal Society University Research Fellowship (UF160152, URF\R\221009). GF acknowledges funding from the Department of Chemistry at Imperial College London.

## Author contributions

J.M.S., S.L., A.A.T., P.W.K.R., and E.F. designed research. J.M.S., S.L., A.A.T., and M.V.G. performed experiments and processed the data. S.L., M.A.K., G.F., and E.F. developed code for data analysis. J.M.S., S.L., A.A.T., and E.F. wrote the manuscript with input from G.F., L.D.M., and P.W.K.R.

## Competing interests

Authors EF, SL, AAT, GF, and LDM, through the Regents of University of California, have filed a patent application in the U.S. Patent and Trademark Office which includes disclosure of inventions described in this manuscript, Provisional Application Serial No. 63/588,142, filed on October 5, 2023, and entitled SINGLE STRANDED RNA MOTIFS FOR IN VITRO CO-TRANSCRIPTIONAL PRODUCTION OF ORTHOGONAL PHASE SEPARATED CONDENSATES. The remaining authors declare no competing interests.
