## [Peer Review File · Nature Communications]

REVIEWER COMMENTS

Reviewer #1 (Remarks to the Author):

This manuscript reports that branched RNA nanostructures named RNA nanostars form self-assembled RNA condensates. In previous studies, the authors have demonstrated nanostar self-assembly using the kissing-loop interaction, a well-known RNA-RNA interaction, in addition to the sticky-end-based nanostar binding well-studied in DNA condensate study. In both cases, it was demonstrated that RNA condensates were formed. Next, using several examples, they showed that the sequence of the kissing-loop interaction and the sequence of the stem portion of the nanostar affected the formation of RNA condensates. They also showed that introducing aptamer sequences into the RNA nanostar specifically recruited fluorescent small molecule (DFHO) or fluorescently labeled peptides into the RNA condensate. Finally, they showed the co-transcriptional RNA condensate formation and demonstrated peptide recruitment as well. Considering the attention paid to condensate formation by DNA nanostars, condensate formation by RNA nanostars will be an equally important technique to be focused on in DNA/RNA nanotechnology. However, overall, the results and discussion are based only on the observation of microscopic images of RNA condensate, and there is no evaluation from a different perspective and little quantitative discussion as to whether molecular self-assembly is occurring as designed, although the reviewer basically agrees with that RNA condensate formation has occurred. Therefore, it has not been verified whether the self-assembly is as rational as the authors' claim, "we demonstrate the rational design of pure RNA condensates from star-shaped RNA motifs." The above points should be further verified if the authors claim the construction is based on a rational design. The following are the reviewer's comments.

(1) p.4: The authors mentioned: "their spherical shape is indicative of a liquid-like state." If they would like to claim the state is actually liquid-like, they should show the fusion dynamics of two condensates or the data of the fluorescence recovery after photobleaching (FRAP). It would be a liquid-like state (the reviewer basically believes), but any scientific data would generally be required.

(2) In Fig. 3C, in the case of 3sv2, changing the stem sequence to stem2 did not work (Fig. 3E), but changing the holding temperature worked (Fig. 3F). The questions on the results are as follows:

(2-1) The difference in length between stem1 (15 nt) and stem2 (16 nt) is only one nucleotide. The authors mention that this result cannot be simply explained by thermodynamics. Can the results be explained by a change in geometry or shape depending on the arm length? Can they use the authors' findings on DNA nanostars (in the previous paper in ACS Nano) to clarify this?

(2-2) In Fig.3E(ii), 3sv2 didn't form coacervate, but can 3sv2 form coacervate if stem3 is used?

(2-3) In Fig. 3E(ii), 3sv2 does not form a coacervate if stem2 is used, but 3sv2 can form a coacervate even if stem1 is used if the hold temperature is decreased to 40°C in Fig. F(i). On the other hand, in Fig. 3E(ii), 3sv4 forms a coacervate if stem2 is used, but 3sv4 cannot form a coacervate even if the hold temperature is lowered to 40°C in F(i). The reviewer feels this is inconsistent. Can you explain this?

(3) It was found that various variants and conditions affected the coacervate formation, but it was not found how the wobble pairs or mismatches affected the formation. Can they explain by comparing the phenomena with physical properties such as stability, structure at base pairing, etc.?

(4) Although the structures were numerically checked by NUPACK, it may be necessary to examine whether the results were achieved actually by the designed and predicted structures, rather than just looking at the coacervate formation under the microscope.

(5) Although the differences in the sequences of stem and KL were able to affect the formation of coacervate, the authors may not be able to claim that they have demonstrated a 'rational' design.

(6) How is the chord length defined for condensates that are connected and form a network-like distribution?

(7) Was the chord length calculated only from one microscope image shown in Fig. 3, or was it calculated from more microscope images? If it was calculated only from one image, the μ _CLD may not reflect important information because the result depended on which part of the image was taken.

(8) In Fig. 5, why were the peptides added after the coacervate formation? Did the peptides inhibit the co-transcriptional formation of coacervate?

(9) Were the coacervates made by the high-yield transcription kit and those done by PURExpress the same? The ones done by PURExpress look larger, and those done by the high-transcription look smaller and many.

(10) Regarding the above, were the RNAs transcribed by the high-yield transcription kit and PURExpress the same? Are there any byproducts of RNAs, and did the byproducts affect the

difference in their appearance? Did the authors check the transcribed RNA by electrophoresis or any other method?

(11) DFHO should be spelled out at the first use: 3,5-difluoro-4-hydroxybenzylidene imidazolinone-2-oxime (DFHO).

(12) The sequence information in the Excel file is not easy to see. If they are written with color highlights, the readability would be improved.

Reviewer #2 (Remarks to the Author):

Membraneless organelles, or biomolecular condensates, play many roles in cell structure and function. Different species of MOs, termed orthogonal condensates, are immiscible and allow for more specific biochemical reactions to be carried out. Therefore, a library of synthetic orthogonal condensates could function as artificial organelles and would be able to create more complex and controllable systems. IDPs and repetitive RNA sequences have been used to create condensates, however they are promiscuous and therefore not orthogonal.

RNA motifs that form condensates by rationally-designed base pairing (RNA nanostars) form condensates, but also can be orthogonal as interactions are controlled. The authors use computational models to predict RNA-RNA interactions to form these RNA nanostars, and they build domains that can bind to small molecules and peptides as well.

Assuming concentration affects coacervate morphology/formation, what is the concentration of RNA in the panels of Figure 2, are they all the same?

I think that it maybe should be mentioned that the initial experiments are using purified RNA, since they are all transcribed by T7 from DNA templates, but it's mentioned the RNA is expressed co-transcriptionally later.

When the temperature hold is first mentioned in the beginning of the results, I think it should be briefly explained. What does the temperature change do for condensate formation?

There is a mention later of the effect of different temperatures on the condensates, but the rationale behind the “melt and hold” protocol should be mentioned earlier.

What was the rationale behind the different variant designs (3sv1-5)?

When I was reading the paper, I was confused what the difference behind the single-stranded RNA nanostars binding client molecules compared to the double stranded ones—is it that the single stranded nanostars can be expressed co-transcriptionally?

It then is mentioned in the discussion that multi- and single-stranded RNA nanostars have different advantages/disadvantages, but I think this should be mentioned earlier in the paper to avoid confusion.

Typo – Figure 3B is mentioned when referencing client molecule binding to coacervates, but I believe it should be 4B.

In Figure 5G, there does appear to be a small amount of binding of TAT peptide to Box B, compared to p22 N peptide to TAR.

It was mentioned that TAT can bind non-specifically to RNA. Why was the streptavidin binding aptamer not instead shown in 5G, if it is known TAT binds nonspecifically?

Overall, the paper takes on a difficult task of creating rationally designed, orthogonal RNA-based condensates that specifically bind cargo.

Condensates are sticky, so creating different condensates that themselves do not mix and bind specific client molecules is impressive. Overall, I found that the experiments carried out were well conducted and analyzed.

However, I think the paper could be improved by further explaining the reasoning behind the conditions/experiments in the results section. I also think explanation of nonspecific TAT binding/why streptavidin was not used in Figure 5G is needed.

Reviewer #3 (Remarks to the Author):

I was asked to review two co-submitted manuscripts sent to different journals with different reviewing rubrics. Using one rubric for both papers makes more sense to me for a simultaneous review, so I've chosen a single rubric and structured my review around its questions. The questions are reprinted below:

-If you recommend publication, please outline, in a paragraph or so, what you consider to be the outstanding features. Do you think the results reported in this manuscript represent a significant advance in the field?

Yes, I recommend publication of this paper (with some minor corrections discussed below). This is because I indeed find the paper to represent a significant advance, specifically in that co-transcriptional creation of a sequence-engineered RNA condensate opens many potential avenues of research in synthetic biology, biomolecular physics, and perhaps even live cell engineering. Compared to engineered peptide condensates, nucleic acids offer more simple programming of particle structure and bonding, and thus of condensate properties, including the orthogonal condensates mentioned here. Further, the creation of a condensate from a single RNA strand (rather than multiple strands) is a clear achievement that simplifies condensate synthesis and design, and avoids potential issues with stoichiometry in multi-strand designs (though potentially introduces other issues, discussed below).

In addition, I find this paper strong in that it tests out various sequence and sticky-end designs. It is excellent that they test out condensate formation in a few different buffer conditions (though note comments about this below). And the authors should be lauded for the detailed methods and controls provided in the paper and supplement, representing adherence to open-science principles that will help insure the impact of their work.

-Has the manuscript flaws that should prohibit its publication? If so, please provide details. In particular, do the authors need to supply additional experimental evidence to back up the claims in their manuscript?

I do not see any major flaws that prohibit publication. Yet there are a variety of minor points that should be addressed before the article is published, which could possibly include an additional experiment (point 3).

1) Generally I am not sure of the utility of the multi-strand RNA nanostars, vs the single-strand. There is not extensive comparison between the two sets of experiments, and so there is no clear conclusion regarding the benefits/drawbacks of the two designs.

2) While the authors do have some discussion of solution ionic conditions, more comment and clarity on this would be helpful. The authors use pre-mixed transcription buffers provided by

companies. Such buffers are typically, in my experience, optimized for transcription in a way that could interfere with, or significantly affect, phase separation and nucleic acid behavior, e.g. including additives such as the multivalent cation spermidine, or very high (~30 mM) magnesium concentrations. Notably, the relevance of such cations for RNA condensate formation was brought into relief by the recent paper from Wadsworth et al. (doi: 10.1038/s41557-023-01353-4), where certain non-basepairing RNAs were driven to phase separate even with only tens of mM of magnesium. The authors should indicate very clearly the composition of their solutions throughout.

3) One control that wasn't apparently done was to directly assay the transcribed RNA, e.g. through gel electrophoresis. I am specifically curious about two things: A) Is there a significant amount of truncated RNA (e.g. abortive transcripts) in the solution? B) Do the RNA molecules form stable multimers through binding between hairpin stem domains? The authors assume that the condensate structure is nanostar-like (with no RNA entanglement, just binding between kissing loops), but don't actually seem to prove this. Notably, there is some literature (see below) indicating multi-hairpin RNAs can form entangled, hybridized, mesh-like condensates through hairpin-stem domain swapping. A specific concern is that the 'melt and hold' protocol could in fact assist in such domain swapping. Giving an estimate of the density of the condensate phase would also help in understanding its structure.

4) As noted further below, there isn't much quantification in this article. A specific question is if the authors can provide some rough estimates of the RNA concentration generated by transcription versus time, and specifically comment on whether it is reasonable to expect enough RNA is generated within 1 hour, per Fig. 5, to create a condensate given, say, estimates of the critical concentration from other nucleic acid condensates

5) I am curious whether condensates sediment and adhere to glass surfaces, and if such adhesion affects the observed morphologies of the condensates. Relatedly, I am curious if handling (pipetting) of the condensates affects the observed morphologies (shear stress during pipetting could affect morphologies, which would be slow to re-equilibrate if the viscosities of the condensates are very high).

6) Comments about the free energy of RNA hybridization might be helpful in the context of the Fig. 2C data—I was surprised that the high UA content sticky ends (nominally weaker binding sequences) led to more gel-like behavior (associated with stronger bonds).

-In the spirit of Occam's razor, do you think the experimental data can be explained using a simpler model or a simpler theory?

The answer to this is basically no, because the paper is phenomenological and doesn't have any serious attempts at quantification/modeling. This isn't generally a weakness—the authors demonstrate many interesting phenomena using the new system they have invented—with one exception: the only attempt at quantification is through the chord length plots, which I do not find very helpful. First, how exactly a chord is defined is not explained in the main text, so it is somewhat confounding jargon. Second, even after knowing what it is (from the supplement), I am not sure

what it is meant to show. As a phase separation phenomenon, the total volume of condensate is a thermodynamically-relevant metric. “chord length” instead is a measure that conflates total volume with the specific kinetic process (e.g. nucleation/growth, or some other process) through which condensate particles form, which makes it hard to use it as the basis for interpretation.

-If the conclusions are not original, it would be very helpful if you could provide relevant references.

The conclusions are clearly original.

-Does this manuscript reference previous literature appropriately?

More discussion of condensates involving RNA hairpins would be useful, in light of my comments above. Work from the Bevilacqua group is particularly relevant; some papers from this group are cited, notably Ref 6 (doi:10.1261/rna.078999.121), but there is no serious consideration of the potential for multivalency to be induced through hairpin domain swapping. I might also point out doi:10.1261/rna.078875.121.

RESPONSE TO REVIEWERS

We thank all Reviewers for volunteering their time and providing insightful feedback that helped us improve our manuscript. All revised text in the manuscript and SI has been highlighted in red. To address their comments, we performed additional experiments including:

1. Fluorescence recovery after photobleaching (FRAP) experiments to evaluate the mobility of nanostars in the dense phase. Fluorescent labeling of RNA molecules was done by substituting 1% of UTP to CY3-labeled UTP in the transcription mix.
2. Experiments tracking the coalescence of individual condensates. These experiments are presented in Fig. 3G and Supplementary Figure 17.
3. Denaturing and native gel electrophoresis experiments to evaluate the length of RNA products and their interactions. These experiments are presented in Supplementary Figures 4, 5 for multi-stranded nanostars and 13, 15, 28, and 30 for single-stranded nanostars.
4. Estimation of the critical concentration for condensation of single-stranded nanostars formed from melt and hold protocol and co-transcriptionally. These experiments are presented in Supplementary Figure 14 and 24 for single-stranded nanostars.
5. Experiments to confirm that peptides do not inhibit co-transcription or condensation of RNA nanostars. We performed experiments by adding both peptides at the beginning of co-transcription and observed no apparent difference from when adding after. These experiments are presented in Fig. 5G and Supplementary Figure 31.
6. Other additional experiments:
 - a. To provide a better understanding of KL interactions, we tested additional variants of single-stranded nanostars, at different hold temperature (Supplementary Figure 12).
 - b. We include orthogonal views of confocal microscopy images, showing sedimentation of condensates on glass slide surface (Supplementary Figure 18).
 - c. We performed a control experiment demonstrating that individual single-stranded nanostars from two-nanostar pairs cannot form condensates during transcription (Supplementary Figure 27).
7. Revision of figures: We updated Figure 3D, panel 3sWT-stem 3, as we initially included an image corresponding to a different variant. The original version of Figure 5G (showing recruitment of peptides added after transcription incubation) has been moved to Supplementary Figure 31. We removed one supplementary figure demonstrating the formation of three-arm multi-stranded nanostars due to lack of reproducibility.

As a result of these changes, the revised manuscript has been substantially improved. We hope that the revised manuscript now meets the very high standards for publication in *Nature Communications*. We thank the editor and Reviewers again for their valuable time and their suggestions for improving our work.

Reviewer #1 (Remarks to the Author):

This manuscript reports that branched RNA nanostructures named RNA nanostars form self-assembled RNA condensates. In previous studies, the authors have demonstrated nanostar self-assembly using the kissing-loop interaction, a well-known RNA-RNA interaction, in addition to the sticky-end-based nanostar binding well-studied in DNA condensate study. In both cases, it was demonstrated that RNA condensates were formed. Next, using several examples, they showed that the sequence of the kissing-loop interaction and the sequence of the stem portion of the nanostar affected the formation of RNA condensates. They also showed that introducing aptamer sequences into the RNA nanostar specifically recruited fluorescent small molecule (DFHO) or fluorescently labeled peptides into the RNA condensate. Finally, they showed the co-transcriptional RNA condensate formation and demonstrated peptide recruitment as well. Considering the attention paid to condensate formation by DNA nanostars, condensate formation by RNA nanostars will be an equally important technique to be focused on in DNA/RNA nanotechnology. However, overall, the results and discussion are based only on the observation of microscopic images of RNA condensate, and there is no evaluation from a different perspective and little quantitative discussion as to whether molecular self-assembly is occurring as designed, although the Reviewer basically agrees with that RNA condensate formation has occurred. Therefore, it has not been verified whether the self-assembly is as rational as the authors' claim, "we demonstrate the rational design of pure RNA condensates from star-shaped RNA motifs." The above points should be further verified if the authors claim the construction is based on a rational design. The following are the Reviewer's comments.

We thank the Reviewer for the encouraging comments and advice.

To improve our quantitative characterization of RNA condensates, we have included new experiments in the revised manuscript, which we believe will address the concerns of the Reviewer.

- 1) We included gel electrophoresis experiments to verify the length and secondary structure of the RNA products, as well as their interaction mechanisms, for selected nanostar variants.*
- 2) The revised manuscript includes more details about the rationale behind the selection of variants (page 5) and more discussion about how nanostars interact.*
- 3) We included Supplementary Table 1 to report the predicted free energy of SE interactions (for multi-stranded nanostars), and we mention it at page 4 of the manuscript.*

- 4) *Revised Supplementary Methods Section 1.1.2 includes the code used for generating stem sequences de novo.*

(1) p.4: The authors mentioned: “their spherical shape is indicative of a liquid-like state.” If they would like to claim the state is actually liquid-like, they should show the fusion dynamics of two condensates or the data of the fluorescence recovery after photobleaching (FRAP). It would be a liquid-like state (the Reviewer basically believes), but any scientific data would generally be required.

We thank the Reviewer for bringing up this point. In the revised manuscript, we included Fluorescence Recovery after Photobleaching (FRAP) experiments for multi-stranded and single-stranded nanostars. We also tracked fusion events for single-stranded nanostars. The results are shown in the new Fig. 3G, Supplementary Figures 7, 16, 17, and 26.

To perform FRAP, we tagged individual RNA molecules by incorporating 1% of CY3-UTP during transcription. Purified RNA samples were annealed following the melt and hold protocol and then imaged. Co-transcribed RNA samples were incubated for three hours after transcription initiation and diluted 10 times with 1x co-transcription buffer prior to imaging, to reduce background fluorescence caused by excessive CY3-UTP.

We tested one multi-stranded nanostar design (variant 4m1) and two single-stranded nanostar designs (3sWT and 3sv2). Fluorescence recovery was monitored for five minutes at the variant hold temperature (40°C or 50°C) and at room temperature.

The condensate variants we tested showed no significant recovery over the observation window, with no difference in recovery between condensates formed from purified RNA (Supplementary Figures 7 and 16) or co-transcriptionally (Supplementary Figure 26). For this reason, we are not reporting estimates of the relaxation time.

Next, by tracking individual droplets via fluorescence microscopy, we examined the fusion kinetics of RNA condensates over the span of several hours, to get an estimate of the ratio of dynamic viscosity over surface tension. We found no significant change of shapes for variant 3sWT even at 50°C. The example condensates of variant 3sv2 shown in Fig. 3G and Supplementary Figure 17 had the relaxation time constant τ to be in the order of about three hours (172 minutes).

In light of these new results, we have revised our comments about the liquid state of condensates. The spherical shape of condensates indicates that either they were liquid at some point during the annealing process, or that their growth time-scale is comparable to their relaxation dynamics.

Our results are overall consistent with the findings of the companion paper (preprint DOI: 10.1101/2023.10.06.561174), under evaluation with a different Nature journal, discussing four-arm single-stranded nanostar designs with different kissing loops.

(2) In Fig. 3C, in the case of 3sv2, changing the stem sequence to stem2 did not work (Fig. 3E), but changing the holding temperature worked (Fig. 3F). The questions on the results are as follows:

To address the comments below we have summarized our answer in a new sentence at page 6 of the manuscript.

(2-1) The difference in length between stem1 (15 nt) and stem2 (16 nt) is only one nucleotide. The authors mention that this result cannot be simply explained by thermodynamics. Can the results be explained by a change in geometry or shape depending on the arm length? Can they use the authors' findings on DNA nanostars (in the previous paper in ACS Nano) to clarify this?

We thank the Reviewer for this comment. Our data indicates that the condensate morphology depends not only on the length but also on the sequence of the stems. Further, based on our data, changing both the sequence of the stem and the sequence of the KL are likely to have "coupled" effects. In other words, a given KL may work with a given stem sequence but not with another, as we found. Stem sequences flanking the KL will affect local stem stability and folding and eventually affect KL interactions. In our previous work (Agarwal et al. ACS nano 2023)¹, we found that longer stems generally improve the growth rate of DNA nanostar condensates. This trend is confirmed by the data in Fig. 3E(i) (3sWT condensates grow faster when longer stems are used); further, stem 1 (15nt) with 3sv4 and 3sv5 generate no condensate while stem 2 (16nt) with the same KLs does (Fig. 3E(ii) and Supplementary Figure 12). However, for 3sv3, nanostar with stem 2 forms smaller condensates than stem 1 (Fig. 3E(ii)), so there are more subtleties about how stem length and sequence affect condensate formation. To elucidate the interplay between KL and stem sequence on the geometry of the nanostars, we are setting up coarse-grained molecular dynamic simulations using oxRNA. Obtaining meaningful and quantitative insights using oxRNA will require a systematic investigation that is outside of the scope of this project.

(2-2) In Fig.3E(ii), 3sv2 didn't form coacervate, but can 3sv2 form coacervate if stem3 is used?

We thank the Reviewer for this question. We designed two new nanostars combining stem3 and KL corresponding to 3sWT and 3sv2, and tested the formation of condensates using our melt and hold protocol. We found that both designs form condensates at 50°C (and thus will also form condensates under a 40°C hold). These new data were added to new Supplementary Figure 12. 3sWT condensates exhibit the morphology of a mixture of very small droplets and large, non-spherical aggregates; 3sv2 condensates form large, spherical droplets. The new experiments are mentioned at page 6 of the revised manuscript.

(2-3) In Fig. 3E(ii), 3sv2 does not form a coacervate if stem2 is used, but 3sv2 can form a coacervate even if stem1 is used if the hold temperature is decreased to 40°C in Fig. F(i). On the other hand, in Fig.3E(ii), 3sv4 forms a coacervate if stem2 is used, but 3sv4 cannot form a coacervate even if the hold temperature is lowered to 40°C in F(i). The Reviewer feels this is inconsistent. Can you explain this?

We thank the Reviewer for this question. First, as discussed in our reply to comment (2-1), we believe that condensation results from the interplay between stem sequence, KL sequence, temperature, etc. which we do not fully understand. Therefore, stem 1 and stem 2 sequences should be examined as distinct designs.

To specifically address this question, we performed additional experiments, and we tested the formation of condensates using stem 2 with KL variants 3sv2 and 3sv4 under a 40°C hold, which we previously had not examined. Condensates formed in both cases (Supplementary Figure 12), confirming that a lower temperature generally has beneficial effects on condensation. We speculate that reasons why variant stem1 3sv4 does not form condensates under both 50°C and 40° C hold include the formation of undesired interaction between stem and KL during annealing, or KL interactions that are too weak at the temperatures we tested. These new experiments are mentioned at page 6 of the revised manuscript.

(3) It was found that various variants and conditions affected the coacervate formation, but it was not found how the wobble pairs or mismatches affected the formation. Can they explain by comparing the phenomena with physical properties such as stability, structure at base pairing, etc.?

Some of our designs include wobble pairs, which are now highlighted with black dots in revised Fig. 3C, while mismatches are highlighted as red dots. Variants 3sv1 (5'-GUGCGC) and 3sv2 (5'-GCGCGU) include two wobble pairs; 3sv3 (5'-GAGCGC), 3sv4 (5'-GCGCUC), and 3sv5 (5'-GCUCGC) include two mismatches of different kind and position. All KL variants generate condensates (stem 1) except 3sv4 under a 40° hold.

In this study, we don't have enough variants to draw conclusions about the influence of wobble pairs vs mismatches in the formation of condensates. We appreciate the Reviewer's advice which is helpful for future work.

*We point out that, as we now clarified in the text, the choice of our variants was based on previous work that tested the dimerization of the HIV kissing loop (reference 36 in the manuscript): Clever, J. L., Wong, M. L. & Parslow, T. G. Requirements for kissing-loop-mediated dimerization of human immunodeficiency virus RNA. *J. Virol.* 70, 5902–5908 (1996).*

KL WT, v1, v3 that were found to dimerize in Clever et al. produce condensates when included in nanostars under a 40°C and 50°C hold; variant v2 only dimerizes with a 40°C hold. We also tested two KL variants that did not dimerize in Clever et al. (corresponding to 3sv4 and 3sv5), and we found 3sv5-stem 1 and 3sv4-stem 2 to form condensates at 40°C.

(4) Although the structures were numerically checked by NUPACK, it may be necessary to examine whether the results were achieved actually by the designed and predicted structures, rather than just looking at the coacervate formation under the microscope.

To address this comment, we included new gel electrophoresis data (denaturing and non-denaturing) in the revised manuscript to prove that predicted structures mediate the interactions.

- a) *Multi-stranded nanostars formed under melt and hold anneal (Supplementary Figure 4 and 5). Denaturing gels illustrate the length distribution of purified RNA products. Native gels show complexes arising when different strands are mixed and annealed. These complexes are associated with bands and smearing that increase when all required strands are present, which is indicative of the formation of networks of variable size, visible under the microscope as condensates.*
- b) *Single-stranded nanostars formed with the melt and hold anneal (Supplementary Figure 13 and 15) and during transcription (Supplementary Figure 28 and 30). Denaturing gels (Supplementary Figure 13 and 30) confirm the length distribution of products, and show the presence of truncated products. Native gels of samples obtained using the melt and hold protocol (Supplementary Figure 15) include 3sWT and 3sv2 nanostars, and show that complexes do not migrate likely due to their size and stability. The replacement of one KL with a polyA sequence (of the same length) results in the emergence of bands that are expected to correspond to multimers forming via stem-stem interactions (no condensates are visible in microscopy images). We also tested a two-nanostar variant, 3s β 1-3s β 2, and found that when only one nanostar is present (either 3s β 1 or 3s β 2 alone), only multimer bands appear, like in the case where one KL is replaced with a polyA sequence. Again, these multimers likely form via stem-stem interactions (no condensates are visible in microscopy images). Native gels of samples obtained with the co-transcriptional protocol (Supplementary Figure 28) confirm that variant 3sv2 forms large complexes that do not migrate in the gel, but replacement of one KL with a polyA sequence results in multimers (no condensates visible in microscopy images). Similarly, when 3s β 1 or 3s β 2 are transcribed together, they form large complexes that do not migrate in the gel, but when they are transcribed alone they only form multimer bands.*

Collectively, these results confirm that KL interactions are the primary drivers of condensate formation.

(5) Although the differences in the sequences of stem and KL were able to affect the formation of coacervate, the authors may not be able to claim that they have demonstrated a ‘rational’ design.

We understand the Reviewer’s point. We agree that our current set of data does not permit the formulation of a complete, predictive model linking nanostar design parameters and condensate properties. However, the nanostar design was formulated following a rational process that includes comparison with previous DNA nanostar designs (for example, Sato et al., 2020²) as well as following the idea that a nanostructured molecule would make it possible to spatially localize bonds (hybridizing sticky ends of KL) and other domains. Following this principle, for example, three bonds should be sufficient for generating condensates; this is indeed what we found. In contrast, RNA condensates obtained via sequence repeats (Jain and Vale, 2017³) require a much higher number of “unstructured” bonds. The architecture of our nanostar also

permits the addition of non-interacting domains such as FLAPs that minimally affect condensation.

In contrast, disruption or elimination of one out of three bonds should suppress condensation, which is what we found by replacing one KL with a polyA sequence.

To address this comment, we have removed the word “rational” from the abstract of the revised manuscript.

Our results are the starting point from which it will be possible to test a variety of nanostar designs.

(6) How is the chord length defined for condensates that are connected and form a network-like distribution?

To address this comment, we have included a brief description of the chord length distribution (CLD) computation in the main text and in the Supplementary Methods section 1.4.1. For all samples, CLDs were computed on binarised epifluorescent microscopic images captured at the optimal focal planes, which provides optimal focusing and represents the overall morphology. In other words, the CLDs in our paper contain information only about the selected focal plane. For each field of view, chords were applied as lines with fixed space (in our case, 1 pixel) intersecting condensates in both x- and y-axis directions, and chord lengths were calculated as the number of pixels of each line and calibrated to micrometers. This was done in the same way for all condensate morphologies, spheres and networks. CLDs provide general information about the length-scale of objects, regardless of their specific shape.

(7) Was the chord length calculated only from one microscope image shown in Fig. 3, or was it calculated from more microscope images? If it was calculated only from one image, the μ _CLD may not reflect important information because the result depended on which part of the image was taken.

All CLDs presented in the paper were computed based on three experimental replicates, and for each replicate we examined 10 fields of view (for multi-stranded nanostar) or 7 fields of view (for single-stranded nanostar). Hundreds to thousands of condensates are used to generate each violin plot (depending on the nanostar design). This information can be found in the Supplementary Materials and Methods Section 1.4.1 for multi-stranded nanostars, and Section 1.4.2 for single-stranded nanostar.

(8) In Fig. 5, why were the peptides added after the coacervate formation? Did the peptides inhibit the co-transcriptional formation of coacervate?

We thank the Reviewer for this comment. To address this comment, we ran a new experiment in which peptides were added at the beginning of transcription (revised Figure. 5G). The addition of peptides at the beginning does not compromise the formation of condensates. The data previously included in Fig. 5 (summarizing experiments in which peptides were added after condensate formation) were moved to Supplementary Figure 31. In the new experiments, we observed slightly stronger non-specific recruitment of the positively charged TAT peptides to

variant 3sv2 including BoxB aptamer, consistent with our results with multi-stranded nanostars (Supplementary Figure 9).

(9) Were the coacervates made by the high-yield transcription kit and those done by PURExpress the same? The ones done by PURExpress look larger, and those done by the high-transcription look smaller and many.

The nanostar variant transcribed with the two kits is the same, but we agree the resulting condensates have different morphologies. The rate of transcription and the different components in the environment (ions, enzymes, amino acids) produce different condensate numbers and morphology. If we compare the yield of condensates after 3 hours, there are more and larger condensates in the transcription-kit sample when compared to the PURExpress sample. We did not monitor the transcription kit products beyond 5 hours. There are large, visible condensates in the PURExpress sample only after 9 hours. We note that the PURExpress system is not designed for high-yield T7 transcription, so its transcription speed is lower when compared with the high-yield T7 transcription kit.

(10) Regarding the above, were the RNAs transcribed by the high-yield transcription kit and PURExpress the same? Are there any byproducts of RNAs, and did the byproducts affect the difference in their appearance? Did the authors check the transcribed RNA by electrophoresis or any other method?

To confirm the size of RNA nanostars in PURExpress, in the revised Supplementary Information File we added gel electrophoresis data of transcription products when using our transcription kit (Supplementary Figure 28) and when using PURExpress (Supplementary Figure 30).

Products of the target size are present along with byproducts in both samples. We note a larger fraction of shorter and elongated transcripts in the PURExpress system. The additional components in PURExpress (ions, enzymes, amino acids), when compared to transcription-only kits, are likely affecting the efficiency of transcription, as well as the appearance of condensates, because macromolecular crowding influences phase separation⁴.

(11) DFHO should be spelled out at the first use: 3,5-difluoro-4-hydroxybenzylidene imidazolinone-2-oxime (DFHO).

We have corrected this in the revised manuscript.

(12) The sequence information in the Excel file is not easy to see. If they are written with color highlights, the readability would be improved.

To address this comment, the Excel file including the sequences has been revised, and different domains are now distinguished by color.

Reviewer #2 (Remarks to the Author):

Membraneless organelles, or biomolecular condensates, play many roles in cell structure and function. Different species of MOs, termed orthogonal condensates, are immiscible and allow for more specific biochemical reactions to be carried out. Therefore, a library of synthetic orthogonal condensates could function as artificial organelles and would be able to create more complex and controllable systems. IDPs and repetitive RNA sequences have been used to create condensates, however they are promiscuous and therefore not orthogonal.

RNA motifs that form condensates by rationally-designed base pairing (RNA nanostars) form condensates, but also can be orthogonal as interactions are controlled. The authors use computational models to predict RNA-RNA interactions to form these RNA nanostars, and they build domains that can bind to small molecules and peptides as well.

We thank the Reviewer for this overall positive evaluation that our paper provided rationally designed RNA nanostars with the ability to interact orthogonally and bind to molecules.

(1) Assuming concentration affects coacervate morphology/formation, what is the concentration of RNA in the panels of Figure 2, are they all the same?

All multi-stranded RNA condensates are annealed and imaged at 5 μ M concentrations. This is mentioned in the Methods section (Condensate preparation) of the manuscript.

(2) I think that it maybe should be mentioned that the initial experiments are using purified RNA, since they are all transcribed by T7 from DNA templates, but it's mentioned the RNA is expressed co-transcriptionally later.

We have modified the manuscript and clarified how RNA was made at the beginning of sections on pages 3 and 5 for multi- and single-stranded nanostars.

(3) When the temperature hold is first mentioned in the beginning of the results, I think it should be briefly explained. What does the temperature change do for condensate formation?

There is a mention later of the effect of different temperatures on the condensates, but the rationale behind the “melt and hold” protocol should be mentioned earlier.

We thank the Reviewer for bringing this to our attention. We have modified the manuscript to explicitly explain the purpose of each step within the “melt and hold” protocol the first time it is mentioned on page 3.

(4) What was the rationale behind the different variant designs (3sv1-5)?

To address this question, we have added a sentence on page 5 of the manuscript. The selection of and naming order of these variants was based on their ability to trigger dimerization of Human Immunodeficiency Virus (HIV) RNA demonstrated in previous literature (Clever, J. L., Wong, M. L. & Parslow, T. G. Requirements for kissing-loop-mediated dimerization of human

immunodeficiency virus RNA. J. Virol. 70, 5902–5908 (1996), reference 36 in the manuscript)⁵. Their probability of dimerization was modulated by introducing mismatches and wobble pairs within the KL sequence. KL that were found to dimerize in by Clever et al. (corresponding to 3sv1, 3sv2, 3sv3) produce condensates when included in nanostars (stem 1) under a 40°C hold. We also tested two KL variants that did not dimerize in reference 32 (corresponding to 3sv4 and 3sv5), and we found 3sv5-stem 1 to form condensates at 40°C.

(5) When I was reading the paper, I was confused what the difference behind the single-stranded RNA nanostars binding client molecules compared to the double stranded ones—is it that the single stranded nanostars can be expressed co-transcriptionally?

It then is mentioned in the discussion that multi- and single-stranded RNA nanostars have different advantages/disadvantages, but I think this should be mentioned earlier in the paper to avoid confusion.

We thank the Reviewer for this feedback. We noticed the mention of a "double-stranded nanostar" in your comments. Our paper reported multi-stranded and single-stranded nanostars; there is no discussion of a double-stranded nanostar. We assume you are referring to the multi-stranded nanostar construct and will address your question accordingly.

The recruitment domains or, in other words, RNA aptamers, are the same across the two designs. However, we started testing nanostar binding to client molecules using multi-stranded nanostars because to include an aptamer it is sufficient to modify only one short strand out of the four strands required. Additionally, the short length of multi-stranded nanostar strands makes DNA template synthesis easier. Appending longer aptamers (e.g. Streptavidin) to single-stranded nanostars causes the template strand length to exceed the manufacturing limit. Meanwhile, the main advantage of using single-stranded nanostars, as the Reviewer mentioned, is that they can be expressed co-transcriptionally. This feature makes it possible for co-transcriptional recruitment as we demonstrated in Fig. 5G. To address this comment we included a new sentence at page 9 of the revised manuscript.

(6) Typo – Figure 3B is mentioned when referencing client molecule binding to coacervates, but I believe it should be 4B.

We regret this oversight and we thank the Reviewer for pointing this out. The typo has been fixed in the revised manuscript and is highlighted in red.

(7) In Figure 5G, there does appear to be a small amount of binding of TAT peptide to Box B, compared to p22 N peptide to TAR.

It was mentioned that TAT can bind non-specifically to RNA. Why was the streptavidin binding aptamer not instead shown in 5G, if it is known TAT binds nonspecifically?

We thank the Reviewer for asking this important question. We agree with the Reviewer that a small portion of TAT peptide can bind non-specifically to RNA due to its positively charged amino acid residue, consistent with results from multi-stranded nanostars (Supplementary

Figure 10). We note that revised Fig. 5G now includes images of condensates produced when peptides are present from the start of transcription (the previous version of Fig. 5G has become Supplementary Figure 31). The non-specific binding of TAT to BoxB aptamer is present also in the new images.

The reason that we didn't use streptavidin-binding aptamer in Fig. 5G is due to DNA synthesis limitations. Our single-stranded nanostar comprises three highly stable hairpin structures, which make T7 polymerase prone to dissociation: for this reason, the templates are difficult to PCR-amplify in house, and they are ordered from Integrated DNA Technologies as full-length template strands. IDT Ultramer synthesis has length limitations (200 bp), and appending streptavidin aptamer to single-stranded nanostar exceeds this limit. Nanostar templates cannot be ordered as G-blocks, because their complexity score exceeds IDT standards. We are working on testing different DNA polymerases and optimizing PCR conditions to overcome these length limitations.

Overall, the paper takes on a difficult task of creating rationally designed, orthogonal RNA-based condensates that specifically bind cargo.

Condensates are sticky, so creating different condensates that themselves do not mix and bind specific client molecules is impressive. Overall, I found that the experiments carried out were well conducted and analyzed.

However, I think the paper could be improved by further explaining the reasoning behind the conditions/experiments in the results section. I also think explanation of nonspecific TAT binding/why streptavidin was not used in Figure 5G is needed.

We thank the Reviewer for the positive feedback and constructive comments. We have addressed the aforementioned questions: the reasoning behind our experimental conditions is discussed in our response to the Reviewer's comments (1), (2), and (3); questions about experimental designs are answered at points (4) and (5). Non-specific binding of TAT and the reasoning for not using streptavidin is responded to in (7). The manuscript and supplementary materials are revised to address these comments.

Reviewer #3 (Remarks to the Author):

I was asked to review two co-submitted manuscripts sent to different journals with different reviewing rubrics. Using one rubric for both papers makes more sense to me for a simultaneous review, so I've chosen a single rubric and structured my review around its questions. The questions are reprinted below:

I recommend publication of this paper (with some minor corrections discussed below). This is because I indeed find the paper to represent a significant advance, specifically in that co-transcriptional creation of a sequence-engineered RNA condensate opens many potential

avenues of research in synthetic biology, biomolecular physics, and perhaps even live cell engineering. Compared to engineered peptide condensates, nucleic acids offer more simple programming of particle structure and bonding, and thus of condensate properties, including the orthogonal condensates mentioned here. Further, the creation of a condensate from a single RNA strand (rather than multiple strands) is a clear achievement that simplifies condensate synthesis and design, and avoids potential issues with stoichiometry in multi-strand designs (though potentially introduces other issues, discussed below).

We appreciate the Reviewer's recognition of the significance of our work in advancing the field of synthetic RNA condensates, particularly that our work offers a platform that can be easily combined with other technologies to solve scientific and practical problems.

In addition, I find this paper strong in that it tests out various sequence and sticky-end designs. It is excellent that they test out condensate formation in a few different buffer conditions (though note comments about this below). And the authors should be lauded for the detailed methods and controls provided in the paper and supplement, representing adherence to open-science principles that will help insure the impact of their work.

We thank the Reviewer for acknowledging our focus on answering fundamental questions and ensuring reproducibility.

I do not see any major flaws that prohibit publication. Yet there are a variety of minor points that should be addressed before the article is published, which could possibly include an additional experiment (point 3).

1) Generally I am not sure of the utility of the multi-strand RNA nanostars, vs the single-strand. There is not extensive comparison between the two sets of experiments, and so there is no clear conclusion regarding the benefits/drawbacks of the two designs.

To address this comment, as well as comment (5) from Reviewer 2, we have added a new sentence at page 9 in the section "Orthogonal RNA condensates can be programmed to recruit guest molecules". In our view, the main advantage of multi-stranded nanostars is that modifying one out of the four strands is sufficient to include an aptamer for molecule recruitment. The short length of strands also makes it easier to be manufactured by vendors. As a result, it is feasible to test new designs with various recruitment domains using multi-stranded nanostars. On the other hand, the secondary structure and longer length of single-stranded nanostars make their DNA template harder to manufacture; some candidate designs (e.g., including a Streptavidin aptamer) could not be synthesized as their template strands exceed manufacturing limits. The primary advantages of single-stranded nanostars is that they can be generated co-transcriptionally and isothermally, making them suitable to be used for potential applications in vivo. On the contrary, forming condensates isothermally from multi-stranded nanostars in vivo is unlikely to work because hybridization of four distinct strands is difficult without thermal annealing, and to obtain a sufficient yield the four strands would have to be transcribed at comparable stoichiometry.

2) While the authors do have some discussion of solution ionic conditions, more comment and clarity on this would be helpful. The authors use pre-mixed transcription buffers provided by companies. Such buffers are typically, in my experience, optimized for transcription in a way that could interfere with, or significantly affect, phase separation and nucleic acid behavior, e.g. including additives such as the multivalent cation spermidine, or very high (~30 mM) magnesium concentrations. Notably, the relevance of such cations for RNA condensate formation was brought into relief by the recent paper from Wadsworth et al. (doi: 10.1038/s41557-023-01353-4), where certain non-basepairing RNAs were driven to phase separate even with only tens of mM of magnesium. The authors should indicate very clearly the composition of their solutions throughout.

We agree with the Reviewer that ionic conditions strongly influence condensate formation and morphologies. Our assembly and transcription buffers were made in house, so their composition is known. PURExpress is the only proprietary buffer we used for which we do not have detailed information about its composition.

To address this comment we have mentioned composition of each buffer in the text (pages 3 and 10) and improved the clarity of the Methods sections.

For the Reviewer's convenience, below is a summary of the buffers we used:

Nomenclature of the buffer in the paper	Condensation in this buffer	Components
Standard assembly buffer	Assembly of purified RNA into condensate (Fig. 2 and 3)	5μM RNA 40 mM HEPES 100 mM KCl 500 mM NaCl
High-yield transcription buffer	Co-transcriptional assembly of condensates (Fig. 5A-D, 5G)	10nM DNA template 40mM of Tris-HCl 10 mM of NaCl 30 mM MgCl₂ 2 mM spermidine 7.5 mM each NTP 10 mM DTT
PURExpress	Co-transcriptional assembly of condensates (Fig. 5E-F)	Provided by the manufacturer.

Regarding the Reviewer's concern about the use of multivalent cations and magnesium, condensates demonstrated in Fig. 2 and Fig. 3 are formed in our assembly buffer which does not include multivalent cations.

Our high-yield transcription buffer for co-transcription experiments contains 2 mM spermidine and 30 mM magnesium, which could promote non-specific condensation akin to Wadsworth's observations. This is unlikely to occur based on our results with:

1) One-nanostar control experiments in which one KL was replaced with a poly-A sequence (compare Fig. 2 C,D and Supplementary Figure 11); no condensates are observed in these variants, although the other 2 KL and the hairpins may still interact.

2) Two-nanostar experiments, in which individual nanostars have non-palindromic KL (e.g., 3s β). In these control experiments, no condensates form when a single nanostar is annealed or transcribed (3s β 1 or 3s β 2). In contrast, condensates grow rapidly when both nanostars are present (e.g., 3s β 1 and 3s β 2), under the same temperature and buffer assembly conditions as in the control (Fig. 5C-D and Supplementary Figure 27).

These two sets of experiments indicate that specific KL binding is the primary driver of phase separation, rather than stem-stem interactions or multivalent cations in the buffer.

3) One control that wasn't apparently done was to directly assay the transcribed RNA, e.g. through gel electrophoresis. I am specifically curious about two things: A) Is there a significant amount of truncated RNA (e.g. abortive transcripts) in the solution? B) Do the RNA molecules form stable multimers through binding between hairpin stem domains? The authors assume that the condensate structure is nanostar-like (with no RNA entanglement, just binding between kissing loops), but don't actually seem to prove this. Notably, there is some literature (see below) indicating multi-hairpin RNAs can form entangled, hybridized, mesh-like condensates through hairpin-stem domain swapping. A specific concern is that the 'melt and hold' protocol could in fact assist in such domain swapping. Giving an estimate of the density of the condensate phase would also help in understanding its structure.

We thank the Reviewer for bringing this to our attention. To address this comment, as well as comment (1) from Reviewer 1, we included new gel electrophoresis data (denaturing and non-denaturing) in the revised manuscript.

A) In the single stranded RNA nanostar denaturing gels there are bands corresponding to abortive transcripts, whose length appears correlated with half the length of a nanostar (Supplementary Figure 13). No major abortive or elongated transcripts were found in the multi-stranded nanostar gels, although we noted some variability in the major product band (Supplementary Figure 3)

B) Native gels of multi-stranded nanostar condensates show smears rather than well-defined bands that would correspond to multimers (Supplementary Figure 4). Native gels of single stranded nanostars do not show smears or multimers, as most products are too large and stable to migrate in the well (Supplementary Figure 13). We observe a small fraction of complexes that are consistent with multimers only in two cases:

1) Single-nanostar variants 3sv2 (palindromic KL) in which one KL is replaced with a polyA sequence (3sv2_1polyA); these multimers could form as nanostars interact with the remaining KL or through their stems.

2) Individual nanostars of the 2-nanostar variants (non-palindromic KL), specifically we tested individual nanostars from the 3s β variant. These multimers could only form from stem-interactions, since KL should not self-interact. However, they are a small fraction of the sample and based on our experiments with the two nanostars system (non-palindromic KL), we exclude that stem-interactions are the main drivers of condensation. When both nanostars are

present, then condensate formation manifests again as thin bands in the well that do not migrate.

For co-transcriptionally produced nanostars (Supplementary Figure 28), the bands in these gels do not significantly differ from those corresponding to the purified and annealed RNA samples.

We found that some variants can yield condensates at 70°C, and we believe that these are cases in which partially melted stems could be the primary mediators of condensation (Fig. 3 F(i) and (ii)).

4) As noted further below, there isn't much quantification in this article. A specific question is if the authors can provide some rough estimates of the RNA concentration generated by transcription versus time, and specifically comment on whether it is reasonable to expect enough RNA is generated within 1 hour, per Fig. 5, to create a condensate given, say, estimates of the critical concentration from other nucleic acid condensates

We thank the Reviewer for this important suggestion. To address this comment we included two new sets of experiments in the revised manuscript.

1) We provide a gel electrophoresis estimation of RNA concentration generated co-transcriptionally versus time (new Supplementary Figure 24). Co-transcription reactions were quenched after 0, 1, 2, 3, 4, and 5 hours by adding DNase I, then loaded onto a denaturing gel. Gel-purified variant 3sv2 at 100 nM was loaded as a reference.

Condensates visible in fluorescence microscopy appear after 1 hour of co-transcription (Fig. 5A) which corresponds to an estimated nanostar concentration under 2 μ M.

2) To estimate the critical concentration for condensation, we formed condensates using column-purified and gel extracted RNA with our melt and hold protocol in our assembly buffer (new Supplementary Figure 14 A). We found that condensates form with as little as 100 nM nanostar concentration. However, the critical concentration appears higher for non-gel extracted RNA, which include truncated products produced during transcription, as an estimated 2 μ M RNA concentration only yields few, very small condensates (Supplementary Figure 14 B).

5) I am curious whether condensates sediment and adhere to glass surfaces, and if such adhesion affects the observed morphologies of the condensates. Relatedly, I am curious if handling (pipetting) of the condensates affects the observed morphologies (shear stress during pipetting could affect morphologies, which would be slow to re-equilibrate if the viscosities of the condensates are very high).

To address the first part of this comment, we imaged condensates using confocal microscopy and provided orthogonal views in Supplementary Figure 18. Variant 3sv2 was annealed following the melt and hold protocol by holding at 40°C. We did not observe significant deformation of droplets from sedimentation or significant wetting of the surface. We attribute this to surface tension too large for gravitational deformation, and no significant affinity for the surface.

To address the second part of this comment, we performed an additional experiment by imaging the same sample after 0, 5, and 10 times of pipetting (2.5 μ L out of the 10 μ L sample), as shown below. We didn't observe any significant change in condensates' morphologies. We would like to note that without pipetting, we observe more condensate accumulation close to the bottom of the tube, probably due to sedimentation. We avoided applying extra shear stress to annealed samples by doing no vortexing and limited pipetting. Based on the condensates' relatively stiff state and the brief handling process, we believe that influence of pipetting to morphological changes are negligible.

6) Comments about the free energy of RNA hybridization might be helpful in the context of the Fig. 2C data—I was surprised that the high UA content sticky ends (nominally weaker binding sequences) led to more gel-like behavior (associated with stronger bonds).

We thank the Reviewer for this important question. We added Supplementary Table 1 summarizing NUPACK predicted free energy of SE interaction for reference. But the network forming cannot be simply explained thermodynamically. In the case of 4m4 in Fig. 2C, either U-A, U-A or U-U, U-U quartets/tetrads may be forming. The literature mentions that the presence of 5'-UA and U-3' flanking nucleotides may form an additional U tetrad, or UA tetrads^{6,7,8} in RNA. These weakly interacting multivalent interactions may cause the formation of gel-like aggregates⁹. We edited the manuscript on Page 4 to include more discussion.

-In the spirit of Occam's razor, do you think the experimental data can be explained using a simpler model or a simpler theory?

The answer to this is basically no, because the paper is phenomenological and doesn't have any serious attempts at quantification/modeling. This isn't generally a weakness—the authors demonstrate many interesting phenomena using the new system they have invented—with one exception: the only attempt at quantification is through the chord length plots, which I do not find very helpful. First, how exactly a chord is defined is not explained in the main text, so it is somewhat confounding jargon. Second, even after knowing what it is (from the supplement), I am not sure what it is meant to show. As a phase separation phenomenon, the total volume of condensate is a thermodynamically-relevant metric. "chord length" instead is a measure that

conflates total volume with the specific kinetic process (e.g. nucleation/growth, or some other process) through which condensate particles form, which makes it hard to use it as the basis for interpretation.

We thank the Reviewer for the encouraging comments, particularly that our study achieved the formation of RNA condensates with a new approach. Our main goal here is to introduce RNA nanostars as a new library for building artificial condensates with controlled behavior, and make readers aware of the modularity and tunability of nanostar condensation. In the future, volume estimation via confocal microscopy will be our priority. Our chord length distribution (CLD) analysis is an expedient approach to estimate the length scale of condensed RNA as well as their kinetics of formation, and we are not attempting to derive thermodynamic information from it. To our knowledge, CLD is suitable for characterizing time-dependent features, such as growth, coalescence, and coarsening^{10,11}. Furthermore, when compared with the published literature where condensate kinetics are reported as the standard deviation of pixel intensity¹², change of total area¹, pixel intensity mapping¹³, etc., we believe that CLD provides more insight into not only size but also shape and spatial organization in a large field of view.

To further address the Reviewer's concern, we modified the manuscript and added a brief introduction to CLD analysis in the revised text. We also added a detailed explanation and a schematic of chord length analysis in Supplementary Methods Section 1.4.1. To provide more thermodynamic insights, we added Supplementary Table 1 to summarize the predicted free energy of SE interactions for reference (multi-stranded designs), which is mentioned in the manuscript on Page 4. We do not attempt to provide estimates of free energy gain for KL interactions, as this cannot be accurately estimated with existing thermodynamic parameters and computational methods.

-If the conclusions are not original, it would be very helpful if you could provide relevant references.

The conclusions are clearly original.

We thank the Reviewer for noting the originality of our work.

-Does this manuscript reference previous literature appropriately?

More discussion of condensates involving RNA hairpins would be useful, in light of my comments above. Work from the Bevilacqua group is particularly relevant; some papers from this group are cited, notably Ref 6 (doi:10.1261/rna.078999.121), but there is no serious consideration of the potential for multivalency to be induced through hairpin domain swapping. I might also point out doi:10.1261/rna.078875.121.

We thank the Reviewer for this suggestion. We discussed the impact of interactions between stems in our paper. In Fig. 3F(ii), we attributed the condensate formation of stem 2, KL 3sWT at 70 °C as "Strikingly, variant 3sWT-stem2 forms condensates even with a 70°C hold, likely due to interactions enabled by partial stem melting (Fig. 3F(i) and (ii))." However, as mentioned in

previous questions, we believe that stem multimerization is not the primary mechanism of condensation for single-stranded nanostars, evidenced by no condensation when only individual nanostars with non-palindromic KL were annealed (Supplementary Figure 20).

To address this comment, we edited the manuscript to include the above-suggested references and added more discussion on this topic on Page 7.

References

1. Agarwal, S., Osmanovic, D., Klocke, M. A. & Franco, E. The Growth Rate of DNA Condensate Droplets Increases with the Size of Participating Subunits. *ACS Nano* **16**, 11842–11851 (2022).
2. Sato, Y., Sakamoto, T. & Takinoue, M. Sequence-based engineering of dynamic functions of micrometer-sized DNA droplets. *Sci Adv* **6**, eaba3471 (2020).
3. Jain, A. & Vale, R. D. RNA phase transitions in repeat expansion disorders. *Nature* **546**, 243–247 (2017).
4. André, A. A. M. & Spruijt, E. Liquid-Liquid Phase Separation in Crowded Environments. *Int. J. Mol. Sci.* **21**, (2020).
5. Clever, J. L., Wong, M. L. & Parslow, T. G. Requirements for kissing-loop-mediated dimerization of human immunodeficiency virus RNA. *J. Virol.* **70**, 5902–5908 (1996).
6. Xiao, C.-D., Ishizuka, T. & Xu, Y. Antiparallel RNA G-quadruplex Formed by Human Telomere RNA Containing 8-Bromoguanosine. *Sci. Rep.* **7**, 6695 (2017).
7. Cheong, C. & Moore, P. B. Solution structure of an unusually stable RNA tetraplex containing G- and U-quartet structures. *Biochemistry* **31**, 8406–8414 (1992).
8. Gu, J., Wang, J. & Leszczynski, J. Hydrogen bonding in 5-bromouracil-adenine-5-bromouracil-adenine (T+AT+A) tetrads. *J. Phys. Chem. B* **108**, 9277–9286 (2004).
9. Falkenberg, C. V., Blinov, M. L. & Loew, L. M. Pleomorphic ensembles: formation of large

- clusters composed of weakly interacting multivalent molecules. *Biophys. J.* **105**, 2451–2460 (2013).
10. Testard, V., Berthier, L. & Kob, W. Influence of the glass transition on the liquid-gas spinodal decomposition. *Phys. Rev. Lett.* **106**, 125702 (2011).
 11. Di Michele, L. *et al.* Multistep kinetic self-assembly of DNA-coated colloids. *Nat. Commun.* **4**, 2007 (2013).
 12. Do, S., Lee, C., Lee, T., Kim, D.-N. & Shin, Y. Engineering DNA-based synthetic condensates with programmable material properties, compositions, and functionalities. *Sci Adv* **8**, eabj1771 (2022).
 13. Muñoz-Gil, G. *et al.* Stochastic particle unbinding modulates growth dynamics and size of transcription factor condensates in living cells. *Proc. Natl. Acad. Sci. U. S. A.* **119**, e2200667119 (2022).

REVIEWERS' COMMENTS

Reviewer #1 (Remarks to the Author):

The reviewer confirmed that the authors improved their manuscript with many additional experiments, which contributed to the quantitative aspects of their manuscript. In addition to the additional experiments, the authors properly answered the reviewers' concerns. Therefore, the reviewer thinks that the revised manuscript can meet the criteria of Nature Communications.

Reviewer #2 (Remarks to the Author):

The authors answered all my questions and addressed all concerns. I think the manuscript is in a great shape now, and it will be a very interesting resource to the community.

Reviewer #3 (Remarks to the Author):

The authors adequately responded to my comments, and, as I stated in the initial review, the results are, overall, very much worth of being published. That said, one minor issue remains:

I do appreciate the author's carrying out follow-up experiments (FRAP and coalescence dynamics quantification) to investigate the insightful comment of Reviewer #1 regarding the dynamics and liquid-like nature of the condensate. However, the presentation of that new data, particularly since the data demonstrates extraordinarily slow dynamics, lacks some context and the paper's conclusion overstates certain aspects. Specifically:

1) In their rebuttal regarding the new dynamics data, the authors state "Our results are overall consistent with the findings of the companion paper" (meaning Fabrini et al., which this referee is also reviewing). This is, as a point of fact, incorrect. The present paper reports inverse capillary velocities around $60 \text{ min}/\mu\text{m} = 3600 \text{ s}/\mu\text{m}$. Fabrini et al report values around $150 \text{ s}/\mu\text{m}$. The values are not at all consistent, being different by more than 10x. That said, I see no change needed to the actual manuscript regarding this.

2) What I think does need to change is the sentence in the conclusion bottom pg 13) that reads "Unlike RNA nanostars, these long repeats are expected to form tangled complexes that require annealing to form gels in vitro, and rely on energy-dissipating cellular machinery to remain in a liquid state". The quantitative data on dynamics of RNA nanostars do not support the implication that they are very liquid at all, so the nanostars are highly unlikely to be more liquid than the long repeat condensates. So this sentence should be removed.

3) The presentation of the dynamic data (middle page 6) lacks any contextualizing comments at all regarding the comparison to other condensates. In fact it is rare in the literature to report a biomolecular condensate with inverse capillary velocity > 100 s/ μ m. I feel this should be noted. I don't think it weakens this paper to point this out; solid-like compartments have their utility, and there are interesting questions to consider in future work regarding why the dynamics are so slow here.

Reviewer #1 (Remarks to the Author):

The reviewer confirmed that the authors improved their manuscript with many additional experiments, which contributed to the quantitative aspects of their manuscript. In addition to the additional experiments, the authors properly answered the reviewers' concerns. Therefore, the reviewer thinks that the revised manuscript can meet the criteria of Nature Communications.

We thank the reviewer for spending time and effort. Your insights and constructive comments have improved our manuscript significantly.

Reviewer #2 (Remarks to the Author):

The authors answered all my questions and addressed all concerns. I think the manuscript is in a great shape now, and it will be a very interesting resource to the community.

We appreciate the Reviewer's positive comments. Again, we would like to thank the reviewer for volunteering the time and effort to provide ideas for enhancing our manuscript.

Reviewer #3 (Remarks to the Author):

The authors adequately responded to my comments, and, as I stated in the initial review, the results are, overall, very much worth of being published. That said, one minor issue remains:

We appreciate the Reviewer's satisfaction with the revision.

I do appreciate the author's carrying out follow-up experiments (FRAP and coalescence dynamics quantification) to investigate the insightful comment of Reviewer #1 regarding the dynamics and liquid-like nature of the condensate. However, the presentation of that new data, particularly since the data demonstrates extraordinarily slow dynamics, lacks some context and the paper's conclusion overstates certain aspects. Specifically:

1) In their rebuttal regarding the new dynamics data, the authors state "Our results are overall consistent with the findings of the companion paper" (meaning Fabrini et al., which this referee is also reviewing). This is, as a point of fact, incorrect. The present paper reports inverse capillary velocities around $60 \text{ min}/\mu\text{m} = 3600 \text{ s}/\mu\text{m}$. Fabrini et al report values around $150 \text{ s}/\mu\text{m}$. The values are not at all consistent, being different by more than 10x. That said, I see no change needed to the actual manuscript regarding this.

The Reviewer is correct, we regret our superficial statement. Our single-stranded nanostar-condensates have slower dynamics than those reported in Fabrini et al. We attribute this different behavior to the difference in kissing-loop interaction strength. Although both designs have 6-base-kissing loops, nanostars used by Fabrini et al. have 4 GC and 2 AU pairs. The designs we used to perform FRAP experiments have 6 GC (for 3sWT) or 4 GC and 2 wobble pairs (for 3sv2). 3sv2 demonstrated a significant increase in molecule mobility compared to 3sWT. We anticipate faster fusion dynamics if we were to change kissing-loop interaction strength by decreasing their GC content.

2) What I think does need to change is the sentence in the conclusion bottom pg 13) that reads "Unlike RNA nanostars, these long repeats are expected to form tangled complexes that require annealing to form gels in vitro, and rely on energy-dissipating cellular machinery to remain in a liquid state". The quantitative data on dynamics of RNA nanostars do not support the implication that they are very liquid at all, so the nanostars are highly unlikely to be more liquid than the long repeat condensates. So this sentence should be removed.

We agree with the Reviewer. We removed the sentence. It is correct however that repeat sequences discussed in Ref. 15 were not demonstrated to form isothermally during transcription.

3) The presentation of the dynamic data (middle page 6) lacks any contextualizing comments at all regarding the comparison to other condensates. In fact it is rare in the literature to report a biomolecular condensate with inverse capillary velocity > 100 s/ μ m. I feel this should be noted. I don't think it weakens this paper to point this out; solid-like compartments have their utility, and there are interesting questions to consider in future work regarding why the dynamics are so slow here.

We thank the Reviewer for this constructive comment. At page 6 we have added a new sentence pointing out that our condensates present very slow dynamics when compared with other reported biomolecular condensates.